# A zebrafish screen reveals Renin-angiotensin system inhibitors as neuroprotective via mitochondrial restoration in dopamine neurons

Gha-Hyun J Kim[1,2], Han Mo[1,3], Harrison Liu[4,5], Zhihao Wu[6], Steven Chen[4,7], Jiashun Zheng[8], Xiang Zhao[1], Daryl Nucum[1], James Shortland[1], Longping Peng[1,9], Mannuel Elepano[10], Benjamin Tang[6,10], Steven Olson[7,10], Nick Paras[10], Hao Li[8], Adam R Renslo[4,7], Michelle R Arkin[4,7], Bo Huang[4,5,11], Bingwei Lu[6], Marina Sirota[12], Su Guo[1,2]*

[1]Department of Bioengineering and Therapeutic Sciences and Programs in BiologicalSciences and Human Genetics, University of California, San Francisco, San Francisco, United States; [2]Graduate Program of Pharmaceutical Sciences and Pharmacogenomics, University of California, San Francisco, San Francisco, United States; [3]Tsinghua-Peking Center for Life Sciences, McGovern Institute for Brain Research, Tsinghua University, Beijing, China; [4]Department of Pharmaceutical Chemistry, University of California, San Francisco, San Francisco, United States; [5]Graduate Program of Bioengineering, University of California, San Francisco, San Francisco, United States; [6]Department of Pathology, Stanford University School of Medicine, Stanford, United States; [7]Small Molecule Discovery Center, University of California, San Francisco, San Francisco, United States; [8]Department of Biochemistry and Biophysics, University of California, San Francisco, San Francisco, United States; [9]Department of Cardiovascular Medicine, Longhua Hospital, Shanghai University of Traditional Chinese Medicine, Shanghai, China; [10]Institute for Neurodegenerative Diseases (IND), UCSF Weill Institute forNeurosciences, University of California, San Francisco, San Francisco, United States; [11]Chan Zuckerberg Biohub, San Francisco, United States; [12]Bakar Computational Health Sciences Institute, University of California, San Francisco, San Francisco, United States

*For correspondence:
su.guo@ucsf.edu

**Abstract** Parkinson's disease (PD) is a common neurodegenerative disorder without effective disease-modifying therapeutics. Here, we establish a chemogenetic dopamine (DA) neuron ablation model in larval zebrafish with mitochondrial dysfunction and robustness suitable for high-content screening. We use this system to conduct an in vivo DA neuron imaging-based chemical screen and identify the Renin-Angiotensin-Aldosterone System (RAAS) inhibitors as significantly neuroprotective. Knockdown of the angiotensin receptor 1 (*agtr1*) in DA neurons reveals a cell-autonomous mechanism of neuroprotection. DA neuron-specific RNA-seq identifies mitochondrial pathway gene expression that is significantly restored by RAAS inhibitor treatment. The neuroprotective effect of RAAS inhibitors is further observed in a zebrafish Gaucher disease model and *Drosophila pink1*-deficient PD model. Finally, examination of clinical data reveals a significant effect of RAAS inhibitors in delaying PD progression. Our findings reveal the therapeutic potential and mechanisms of targeting the RAAS pathway for neuroprotection and demonstrate a salient approach that bridges basic science to translational medicine.

**eLife digest** Parkinson's disease is caused by the slow death and deterioration of brain cells, in particular of the neurons that produce a chemical messenger known as dopamine. Certain drugs can mitigate the resulting drop in dopamine levels and help to manage symptoms, but they cause dangerous side-effects. There is no treatment that can slow down or halt the progress of the condition, which affects 0.3% of the population globally.

Many factors, both genetic and environmental, contribute to the emergence of Parkinson's disease. For example, dysfunction of the mitochondria, the internal structures that power up cells, is a known mechanism associated with the death of dopamine-producing neurons.

Zebrafish are tiny fish which can be used to study Parkinson's disease, as they are easy to manipulate in the lab and share many characteristics with humans. In particular, they can be helpful to test the effects of various potential drugs on the condition.

Here, Kim et al. established a new zebrafish model in which dopamine-producing brain cells die due to their mitochondria not working properly; they then used this assay to assess the impact of 1,403 different chemicals on the integrity of these cells. A group of molecules called renin-angiotensin-aldosterone (RAAS) inhibitors was shown to protect dopamine-producing neurons and stopped them from dying as often. These are already used to treat high blood pressure as they help to dilate blood vessels. In the brain, however, RAAS worked by restoring certain mitochondrial processes.

Kim et al. then investigated whether these results are relevant in other, broader contexts. They were able to show that RAAS inhibitors have the same effect in other animals, and that Parkinson's disease often progresses more slowly in patients that already take these drugs for high blood pressure.

Taken together, these findings therefore suggest that RAAS inhibitors may be useful to treat Parkinson's disease, as well as other brain illnesses that emerge because of mitochondria not working properly. Clinical studies and new ways to improve these drugs are needed to further investigate and capitalize on these potential benefits.

## Introduction

As the most common movement disorder and the second most common neurodegenerative disorder, Parkinson's disease (PD) affects 0.3 % of the general population, with a majority of cases being sporadic (*Obeso, 2017*). The hallmark of PD is a selective vulnerability of substantia nigra dopamine (DA) neurons among others (*Gonzalez-Rodriguez et al., 2020*), accompanied by both motor and non-motor features, as well as prominent Lewy body pathology and mitochondrial dysfunction (*Bose and Beal, 2016*; *Tarakad and Jankovic, 2020*). As one of the world's fastest growing neurological disorders, the economic cost of PD is estimated to be at least $51.9 billion a year in the United States (*Yang, 2020*).

The current drug therapies available for PD provide only symptomatic relief by enhancing the dopaminergic action, decreasing metabolism of DA, or replacing the natural form of DA with exogenous drugs (*Haddad et al., 2017*). While it provides benefits in improving motor symptoms in the short term, chronic therapy can result in motor fluctuation and dyskinesia. Furthermore, many patients experience a wearing-off effect where levodopa loses its efficacy even with dosing adjustments. Despite potentially promising efforts (*Dawson and Dawson, 2019*), there is yet to be a therapy that can halt or slow down disease progression.

In recent years, the drug discovery pipeline for neurological diseases has been stagnant, with a phase I to approval rate being only 8.4 % from 2006 to 2015 (*Wong et al., 2019*) which is lower than the average approval rate of all therapeutic indications. One of the reasons for such lack of success can be attributed to the fact that conventional approach of target-based drug discovery is difficult in the setting of neurological diseases because of the complex etiology and biological pathways involved. Phenotypic screening provides a promising opportunity. In particular, the whole organism-based drug discovery has been successfully applied to model organisms (*Szabo, 2017*). Larval zebrafish, being a vertebrate that can fit in 96-well plates, offers multiple advantages including genomic and anatomical conservation in addition to high throughput capabilities (*Guo, 2009*). Screens have uncovered therapeutic leads that are currently in clinical testing and/or shed light on biological mechanisms (*Macrae and Peterson, 2015*; *Zon and Peterson, 2005*; *Baraban et al., 2013*; *Matsuda, 2018*).

Here, we report a DA neuron-based neuroprotective drug discovery pipeline for PD, from assay development to small molecule screening, hit target validation, mechanisms, and ultimately, evolutionary conservation of neuroprotection across species including humans. Due to the late onset and variable phenotypic expressivity of genetic PD models and the highly toxic nature of neurotoxins (e.g. MPTP) to researchers, no good in vivo assay systems exist that are suitable for high content screening of neuro-protectives. We therefore first established an inducible chemogenetic DA neuron ablation model in larval zebrafish. This model expressed the *E. coli* nitroreductase (NTR) controlled by the promoter of tyrosine hydroxylase (*th*), a rate-limiting enzyme in DA synthesis. Addition of the commonly used and safe-to-handle antibiotic, metronidazole (MTZ), caused robust DA neuron loss. By showing that DA neuron loss is preceded by mitochondrial DNA damage and ensuing mitochondrial dysfunction, we demonstrated the validity and relevance of this model to PD for small molecule screening purpose. Using this model to screen >1400 bioactive small molecules, we uncovered a series of compounds that protected against DA neuron loss by inhibiting different proteins in the renin-angiotensin-aldosterone system (RAAS), a pathway classically known for regulating vasoconstriction and water homeostasis (*Bader, 2010*). Genetic validation and molecular characterization revealed that the angiotensin receptor 1 (AGTR1) acted cell autonomously in DA neurons, the inhibition of which restored the expression of mitochondrial pathway genes disrupted by neurotoxic insults. Furthermore, we showed that RAAS inhibitors were neuroprotective in a zebrafish 1-methyl-4-phenyl-1,2,3,6-tetrahydropyridine (MPTP) model and a zebrafish model of Gaucher disease, a lysosomal storage disorder with strong comorbidity of PD (*Riboldi and Di Fonzo, 2019*). The AGTR1 inhibitor olmesartan was also protective in a *Drosophila pink1*-deficient PD model (*Yang et al., 2006*). Finally, utilizing the Parkinson's Progression Marker Initiative (PPMI) database (*Marek, 2018*), we performed a clinical informatics analysis to uncover that RAAS inhibitors significantly slowed down PD progression. Together, our results delineate a powerful approach for neuroprotective small molecule drug discovery that leverages whole organism screening and cross-species validation.

## Results

### The Nitroreductase-Metronidazole (NTR-MTZ) chemogenetic DA neuron ablation model shows mitochondrial damage prior to neuronal loss and is scalable for DA neuron imaging-based small molecule screening

Genetic PD models in rodents generally have weak, variable, and late onset degeneration phenotypes (*Dawson et al., 2010*). Modeling neurodegeneration in zebrafish is a promising approach (*Paquet, 2009*; *Flinn, 2013*; *Ana Lopez, 2017*; *Zhang, 2017*), but neuroprotective screening based on direct imaging of DA neuron integrity has not been reported, in large part because high content screening needs assays that are sufficiently robust, sensitive, and scalable. Neurotoxins such as MPTP are highly toxic to experimenters and not scalable. We have therefore used an inducible chemogenetic DA neuron ablation model, employing the nitroreductase-metronidazole (NTR-MTZ) system (*Williams, 2015*; *Pisharath and Parsons, 2009*; *Curado et al., 2008*): NTR was expressed as a transgene in tyrosine hydroxylase (TH[+]) DA neurons to convert the pro-drug MTZ (a commonly used antibiotic) to the toxic nitroso radical form in vivo. Twenty-four hours (hrs) after adding MTZ to 5 days post fertilization (dpf) larval zebrafish, we observed at 6 dpf robust DA neuronal loss in the ventral forebrain region (*Figure 1A*), the homologous group to mammalian substantia nigDA (*Rink and Wullimann, 2001*). The specificity of ventral forebrain DA neuron labeling in this transgenic line has been previously validated (*Liu, 2016*).

Although the NTR-MTZ system has been used for cell ablation and noted to induce apoptotic cell death (*Chen et al., 2011*), the underlying mechanisms of cell death are not well understood. Previous reports suggest that MTZ as an antibiotic targets bacterial DNA (*Edwards, 1979*). In vertebrate cells, two organelles containing DNA are the nucleus and the mitochondria. A semi-quantitative PCR assay based on the notion that DNA lesions block DNA polymerase progression (*Furda et al., 2012*; *Yakes and Van Houten, 1997*), was used to detect nuclear and mitochondrial DNA integrity. Total DNAs were extracted from DA neurons in DMSO control and 4.5 mM MTZ-treated transgenic larvae (8 hr after MTZ treatment, when DA neurons remain morphologically intact). Anterior brains (rostral to the mid-hindbrain boundary) were dissected and acutely dissociated. The mCherry[+] DA neurons were

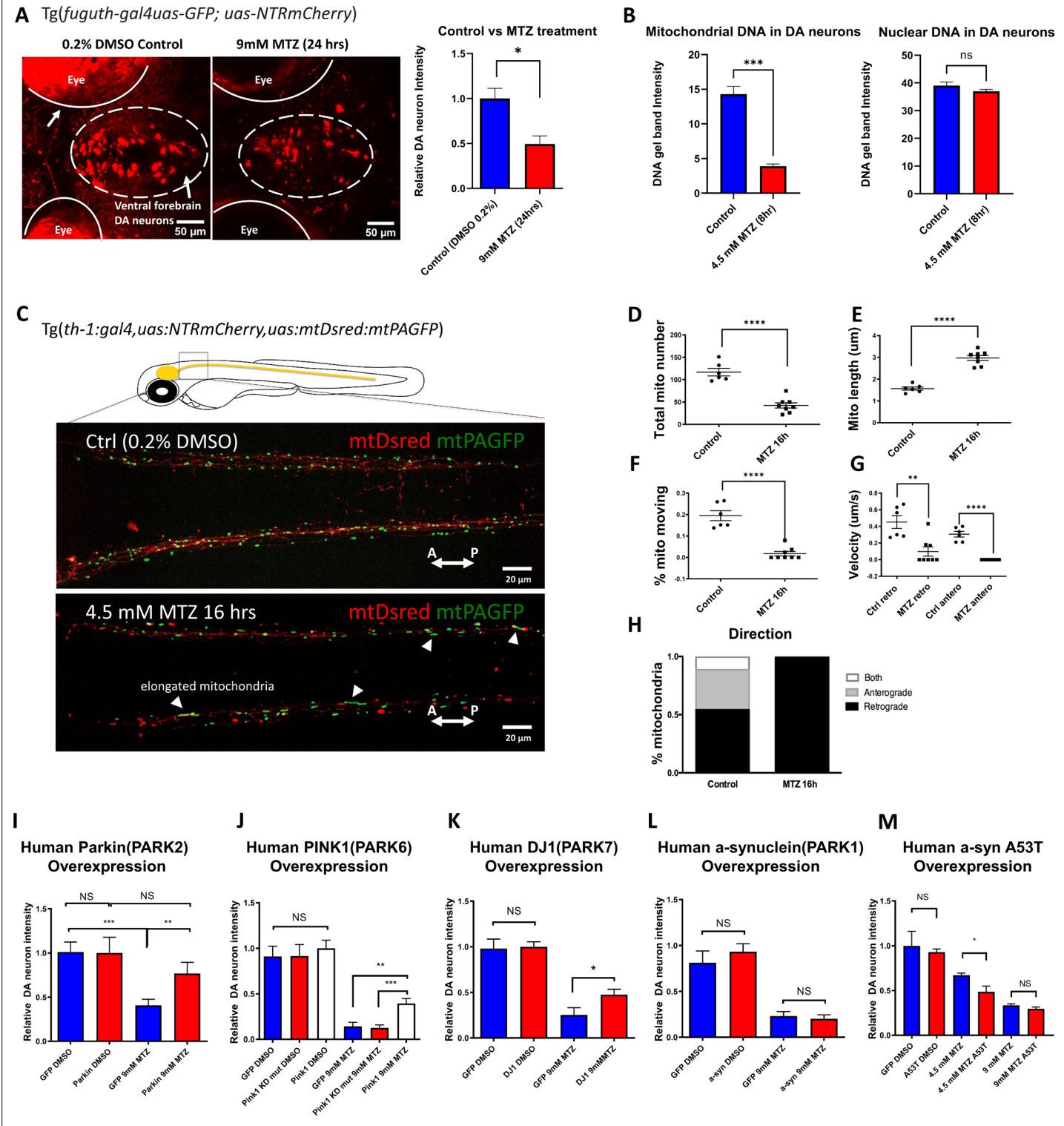

**Figure 1.** The zebrafish NTR-MTZ chemo-genetic DA neuron ablation model suffers from mitochondrial dysfunction that can be counteracted by PD-associated mitochondrial quality control gene activity. (**A**) Confocal images of ventral forebrain DA neurons in 0.2 % DMSO control and 9 mM MTZ-treated 6 days post fertilization (dpf) transgenic larval zebrafish brains show significant difference in normalized fluorescent intensity (n = 10; p < 0.05, unpaired t test). The red fluorescence in the eyes is due to pigment-derived autofluorescence. (**B**) Long-range PCR of mitochondrial DNA versus nuclear DNA products using ventral forebrain DA neurons from control and MTZ-treated 6 dpf larval zebrafish brains anterior to the mid-hindbrain boundary (4.5 mM, 8 hrs) (n = 4 pools of 25 larval brains per pool; p < 0.01, unpaired t test). (**C**) Live confocal imaging of mitochondrial dynamics with mitochondria-targeted DsRed and mitochondria-targeted photoactivatable GFP in 5dpf larvae treated with 0.2 % DMSO (control) or 4.5 mM MTZ for

*Figure 1 continued on next page*

*Figure 1 continued*

16 hr. Arrowheads point to the elongated appearance of mitochondria in DA axons of MTZ-treated animals. (**D–H**) Analysis of mitochondrial dynamics including total mitochondrial count, length, % moving, velocity, and direction of movement between control and MTZ-treated samples (n = 8–10; \*\*p < 0.01, \*\*\*\*p < 0.0001, unpaired t test). (**I–M**) Overexpression of PD-associated human genes including PARK2, PARK6, PARK7, PARK1, and associated mutant forms. mRNAs were microinjected into one-cell stage transgenic embryos and treated with 4.5 or 9 mM MTZ at 30hpf for 24 hrs to determine the neuroprotective effect of experimental conditions compared to control GFP-encoding mRNA injection (n = 10–12; \*p < 0.05, \*\*p < 0.01, \*\*\*p < 0.001, unpaired t test).

The online version of this article includes the following video and figure supplement(s) for figure 1:

**Source data 1.** Dopamine neuron intensity quantification and statistical tests.

**Source data 2.** Labeled mitochondrial DNA damage gel.

**Source data 3.** Labeled nuclear DNA damage gel.

**Source data 4.** Raw mitochondria DNA damage gel.

**Source data 5.** Raw nuclear DNA damage gel.

**Figure 1—video 1.** Movies showing mitochondrial dynamics in DA neuronal axons.
https://elifesciences.org/articles/69795/figures#fig1video1

**Figure 1—video 2.** Movies showing mitochondrial dynamics in DA neuronal axons.
https://elifesciences.org/articles/69795/figures#fig1video2

manually aspirated under a fluorescent microscope. PCR of equally long-length (~10 kb) nuclear or mitochondrial DNA products was carried out using primers as previously described (*Furda et al., 2012*; *Yakes and Van Houten, 1997*). This data uncovered a significant damage of mitochondrial DNA but not nuclear DNA in MTZ-exposed individuals (*Figure 1B*). The nuclear DNA was unaffected possibly due to protection by nucleosomes; this observation also supports the notion that the time of our DNA integrity assessment precedes that of overt neurodegeneration.

We next performed in vivo time-lapse imaging, which uncovered mitochondrial dysfunction in morphologically intact DA neurons after MTZ treatment. These include reduced mitochondrial number, increased mitochondrial length, decreased motility and velocity. Interestingly, in MTZ-treated DA neurons, mobile mitochondria moved exclusively in retrograde direction toward neuronal soma, suggesting that they are targeted for repair and/or mitophagy (*Figure 1C–H*; *Figure 1—videos 1; 2*). Together, mitochondrial defects are observed prior to DA neurodegeneration in the NTR-MTZ model, suggesting that mitochondrial DNA damage and ensuing mitochondrial dysfunction is likely a cause rather than a consequence of DA neuron degeneration.

Given the observed mitochondrial deficits prior to neurodegeneration in the NTR-MTZ model, we next wondered whether enhancing the activity of genes functioning in mitochondrial quality control would protect against neurodegeneration in the model. Homozygous *parkin* mutations account for the majority of early onset autosomal recessive PD (*Kitada, 1998*). The *parkin* gene encodes a conserved E3 ubiquitin ligase that promotes mitochondrial quality control (*Pickrell and Youle, 2015*). The zebrafish Parkin protein is approximately 80 % identical to the human counterpart in functionally relevant domains. We therefore synthesized mRNAs encoding full-length human *parkin* gene or EGFP (control) and micro-injected them into one-cell stage *Tg[fuguth:gal4-uas:GFP; uas-NTRmCherry]* embryos. Expressivity of microinjected mRNAs was verified by observing the GFP fluorescent reporter following *egfp* mRNA injection. Because of the short-lived nature of all microinjected mRNAs, we treated control or *parkin* mRNA-injected embryos with MTZ at an earlier stage, 30 hr post fertilization (hpf), and subsequently imaged for mCherry at 50 hpf (rather than treatment at 5 dpf and imaging at 6 dpf as shown in *Figure 1A*). This regimen of MTZ administration similarly led to DA neuron loss. Increased expression of Parkin significantly protected DA neurons (*Figure 1I*). Moreover, increased expression of *pink1* and *DJ-1*, two other genes associated with mitochondrial quality control (*Pickrell and Youle, 2015*), was also neuro-protective (*Figure 1J–K*). In the case of *pink1*, the kinase dead mutant form failed to show protective effects, further validating the specificity of our assay system (*Figure 1J*). Human α-synuclein (either the wildtype or the A53T mutant form), associated with a dominant form of PD (*Polymeropoulos, 1997*), did not show any protection under the 9 mM MTZ treatment condition; mis-expression of the A53T mutant form of α-synuclein significantly worsened DA neuron integrity under the milder 4.5 mM MTZ treatment condition (*Figure 1L–M*). This is consistent with the toxic nature of α-synuclein when mis-expressed. These results suggest that DA neuron

degeneration in the NTR-MTZ model can be alleviated by enhancing the activity of mitochondrial quality control genes and aggravated by the activity of A53T mutant form of α-synuclein.

## A whole organism DA neuron-imaging based chemical screen identifies inhibitors of RAAS signaling as neuroprotective

We have previously described a high throughput in vivo brain imaging method in zebrafish larvae (*Liu, 2016*). Using this method and the established DA neuron ablation model as described above, we screened >1400 bioactive compounds (SelleckChem) that are part of the UCSF Small Molecule Discovery Center (SMDC) bioactive screening set. Many have validated biological and pharmacological activities, with demonstrated safety and effectiveness in preclinical and clinical research, and some are FDA-approved therapeutics. A dual-flashlight plot of Brain Health Score (BHS) and strictly standardized mean difference (SSMD) score were generated as previously described (*Liu, 2016*), in order to quantitatively document the effects of each compound on DA neuron integrity (*Figure 2A*). In addition to the DMSO-positive control and MTZ-only negative control, N-acetyl cysteine (NAC) was also used as a positive control. NAC is an over-the-counter available supplement that works primarily by restoring body's natural antioxidant glutathione for proposed DA improvement in PD (*Monti, 2019*). In our screen, NAC had an SSMD score of 0.826. Around 100 out of 1403 compounds had the same or better SSMD scores than NAC (ranged from 0.826 to 6.832). Validation of these candidate hit compounds is ongoing. Among them, compounds that target different components of the renin-angiotensin-aldosterone system (RAAS) in particular caught our attention: First, compounds that target three different components of the RAAS signaling pathway (*Figure 2—figure supplement 1*) were all among the top 100 candidate hits [e.g. Olmesartan (angiotensin receptor 1-AGTR1 inhibitor) with the SSMD scores of 1.649 - ranked 14, aliskiren (renin inhibitor)'s SSMD score was 1.540 - ranked 20, imidapril (ACE inhibitor)'s SSMD score was 0.938 - ranked 69]. Additionally, a Wilcoxon rank-sum test comparing all 13 RAAS pathway inhibitors from the primary screen with the entire screening library uncovered a significantly higher SSMD score for RAAS Inhibitors (*Figure 2B*). Multiple hits targeting the same pathway thus built strong confidence in these compounds and in the involvement of RAAS pathway in DA neuron degeneration. Furthermore, literature search has uncovered reports of RAAS pathway inhibition for neuroprotection against PD and Alzheimer's disease (AD) (*Grammatopoulos, 2007*; *Muñoz, 2006*; *Nelson et al., 2013*). However, given the well-established role of these inhibitors in vascular remodeling and blood pressure control, it is conventionally thought that the neuroprotective effects of RAAS inhibition are due to their vascular actions, despite that RAAS pathway expression is detected in the CNS (*Wright and Harding, 2011*). It therefore remains unclear how RAAS inhibition might be neuroprotective. Finally, since RAAS inhibitors are commonly used anti-hypertensives, it offers an opportunity to investigate their role in PD patients via retrospective clinical data analysis. Taken together, we have therefore chosen RAAS inhibitors for in-depth analysis in this study.

We next performed secondary hit validation for olmesartan, captopril (another ACE inhibitor similar to imidapril), and aliskiren. all of them showed significant DA neuron neuroprotection, either by automated fluorescence intensity quantification (*Figure 2C–D*) or blinded manual neuronal counting (*Figure 2—figure supplement 2*). For dose-response curves of these compounds, we implemented dual imaging (imaging both before and after MTZ treatment) to further control for possible individual variability in DA neuron fluorescent intensity (*Figure 2—figure supplement 2B,D*).

With PD being highly associated with motor symptoms and occurring mostly in adult populations, we next determined whether RAAS inhibitors are capable of restoring motor function in adult *Tg[fuguth:gal4-uas:GFP; uas-NTRmCherry]* zebrafish treated with MTZ. Prolonged treatment of MTZ with or without RAAS inhibitors was performed over the period of two weeks in adult transgenic zebrafish, accompanied by locomotor behavioral tracking (*Figure 2E*). As a positive control, we used levodopa, a gold standard symptomatic drug that can restore motor function in PD patients by increasing DA release from surviving neurons. Compared to vehicle controls, MTZ-treated animals showed a progressive decline of locomotive ability for the first 5 days post MTZ treatment and then reached steady low levels. Co-administration of levodopa 1 day after MTZ did not prevent initial locomotor decline, but was able to subsequently restore locomotor function, and interestingly, a hyper-locomotor state was reached at Day 14. The chronic use of levodopa leading to hyperactivity is previously reported in mouse studies (*Gellhaar et al., 2015*) also, uncontrolled involuntary muscle

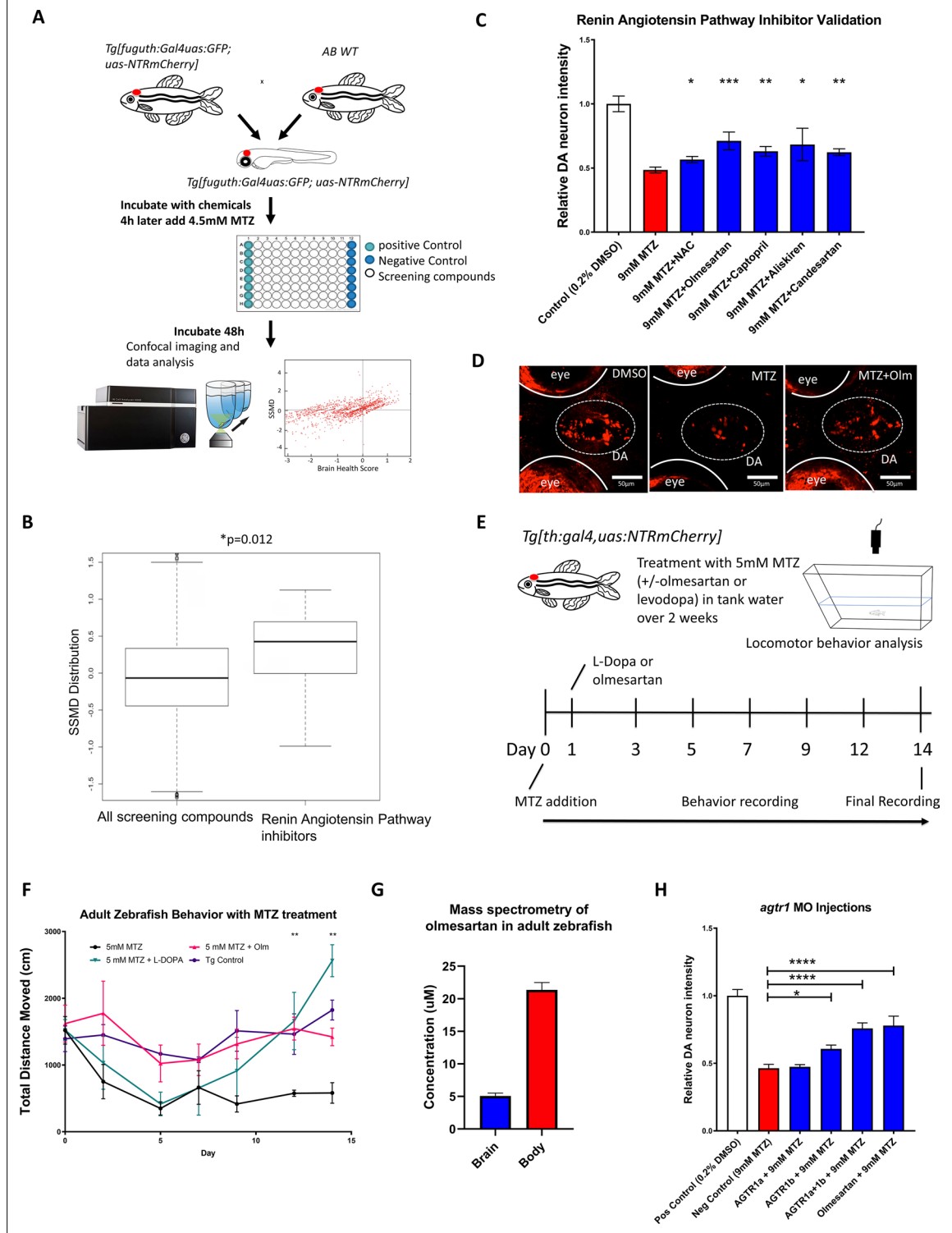

**Figure 2.** A high throughput in vivo imaging-based chemical screen uncovers the neuroprotective effects of inhibiting the renin-angiotensin (RAAS) pathway. (**A**) A flow chart outlining the screening pipeline. 5dpf transgenic larvae expressing *Tg[fuguth:gal4-uas:GFP;uas:NTRmCherry]* were arranged in glass bottom 96-well plates and treated with MTZ (4.5 mM, 48 hrs) along with each of the 1403 bioactive compounds (n = 3 per screening compound). The dual flashlight plot of Brain Health Score (BHS) and Strictly Standardized Mean Difference (SSMD) score was used to quantify the neuroprotective effects of all compounds in the screen. (**B**) Wilcoxon rank sum test was performed to compare data of all 1403 compounds with those representing RAAS inhibitors (n = 13) in the screened compound set, revealing a significantly higher SSMD score distribution in the RAAS inhibitor group (p = 0.012, Wilcoxon rank sum test). (**C**) Secondary hit validation. To obtain more precise data, before and after treatment imaging was carried out for each larva

*Figure 2 continued on next page*

*Figure 2 continued*

embedded in agarose and a treatment regimen with 9 mM MTZ for 24 hr was used. Compounds including the RAAS inhibitors and the N-acetyl cysteine (NAC) control compound were tested at 10 µM with increased sample size (n = 40 per group; *p < 0.05, **p < 0.01, ***p < 0.001, unpaired t test). (**D**) Confocal images of ventral forebrain DA neurons. Positive control (0.2 % DMSO), negative control (9 mM MTZ), and 9 mM MTZ +10 µM olmesartan following 24 hrs of treatment. (**E**) Schematic of the chronic drug treatment and behavior test for adult zebrafish. (**F**) Quantification of total distance traveled across 5 min recording in the home tank for adult zebrafish treated with 0.2 % DMSO (positive control), 5 mM MTZ (negative control), 5 mM MTZ +10 mM levodopa, and 5 mM MTZ +10 µM olmesartan (with daily change of drug solutions after behavioral recording). Distance recordings were conducted for baseline, 3, 6, 9, 12, and 14 days. ANOVA and post-hoc Tukey test showed significant difference in 12 and 14 days for the MTZ versus MTZ+ olmesartan-treated groups. [n = 6 (three males,  three females) for MTZ and MTZ+ Olm, n = 4 (two males,  two females) for DMSO control and levodopa; p < 0.01, one-way ANOVA post-hoc Tukey's test]. (**G**) Mass spectrometry data of adult zebrafish homogenized brain versus body samples after 14 days of chronic treatment with Olmesartan (n = 6,  three males and  three females). (**H**) Quantification of relative fluorescent intensity of DA neurons at 6 dpf in positive control (0.2 % DMSO), negative control (9 mM MTZ, 24 hr from 5 dpf to 6 dpf), *agtr1a* morphant +9 mM MTZ, *agtr1b* morphant +9 mM MTZ, agtr1a/agtr1b double morphant +9 mM MTZ, and 10 µM olmesartan +9 mM MTZ (n = 10–12; *p < 0.05, ***p < 0.001, unpaired t test).

The online version of this article includes the following figure supplement(s) for figure 2:

**Source data 1.** Secondary hit validation for RAAS compounds.

**Source data 2.** Mass spectrometry data and adult behavior.

**Figure supplement 1.** Overview of the classically known renin angiotensin pathway and the inhibitors identified from our high throughput screen.

**Figure supplement 2.** Manual validation and dose response studies of RAAS inhibitors.

**Figure supplement 2—source data 1.** Dose response quantification for captopril, aliskiren, olmesartan.

**Figure supplement 3.** Agtr1a and Agtr1b morpholino phenotypes and western blot validation.

**Figure supplement 3—source data 1.** AGTR1 western blot labeled.

**Figure supplement 3—source data 2.** Beta actin western blot labeled.

**Figure supplement 3—source data 3.** AGTR1 western blot raw.

**Figure supplement 3—source data 4.** Beta actin western blot raw.

**Figure supplement 4.** Olmesartan shows neuroprotective effects post neuronal injury in the NTR-MTZ DA neuron ablation model.

---

movement such as dyskinesia is a common side effect of chronic levodopa use in humans (*Espay, 2018*). In contrast to levodopa, administration of the RAAS inhibitor olmesartan 1 day after MTZ fully ameliorated locomotor defects (*Figure 2F*).

The bioavailability of RAAS inhibitors in the brain and their ability to cross the blood brain barrier (BBB) are generally poor in humans (*Michel et al., 2013*), but it is not known whether olmesartan enters the brain and the conversion of the pro-drug to its active carboxylate form takes place in zebrafish. We therefore performed mass spectrometry after 14 -day treatment of adult zebrafish with olmesartan medoxomil (the pro-drug form of olmesartan). The presence of active olmesartan was detected in the adult zebrafish brain (*Figure 2G*). In addition to locomotor improvements, significant DA neuron protection was also observed in the adult brain setting (*Figure 2—figure supplement 2E-F*).

To verify whether the chemical inhibitors indeed target RAAS signaling components to exert their neuroprotective effects in zebrafish, we knocked down the angiotensin receptor 1 (*agtr1*) gene activity. Two genes (*agtr1a* and *agtr1b*) encode *agtr1* in vertebrates including zebrafish. We designed morpholino (MO) antisense oligonucleotides that inhibited protein translation (ATG MO) (*Figure 2—figure supplement 3A*), and micro-injected them into  one-cell stage embryos. Using an *agtr1* antibody, we verified the protein knockdown in *agtr1a and 1b* double morphants at 6 dpf (*Figure 2—figure supplement 3B*). The morphants appeared morphologically normal. At 5 dpf, control and morphants were treated with 9 mM MTZ for 24 hr. Significant DA neuron protection was observed in *agtr1b* and *agtr1a&1b* double morphants, the extent of which was comparable to olmesartan treatment (*Figure 2H*, *Figure 2—figure supplement 3C*). These data suggest that inhibition of *agtr1* protects against DA neuron degeneration.

In our primary screen and secondary validations, prophylactic dosing was used (i.e. screening compounds were added earlier than MTZ). In order to determine whether the neuroprotective benefits of RAAS inhibitors can be observed after MTZ treatment (i.e. mimicking therapeutic dosing), we carried out MTZ ablation 8 hrs or 24 hrs prior to administering olmesartan (*Figure 2—figure supplement 4A*). In both cases, olmesartan showed significant neuroprotection, suggesting that RAAS

inhibitors can be beneficial even after the onset of neurotoxic insults (*Figure 2—figure supplement 4B*).

## *agtr1* inhibition in DA neurons is neuroprotective

The RAAS pathway as a peptidergic system is composed of ligands and G-protein-coupled receptors (GPCRs) classically known to regulate blood pressure and salt retention (*Bader, 2010*). RAAS inhibitors are widely used drugs for treating high blood pressure. In recent years, RAAS signaling expression is detected outside vasculature and in the central nervous system (*Wright and Harding, 2011*). To understand the potential contribution of neuronal RAAS to DA neuron degeneration, we first performed qPCR on DA neurons purified by Fluorescence-Activated Cell Sorting (FACS) from the anterior brains of 5 day old *Tg[fuguth:gal4-uas:GFP; uas:NTRmCherry]* larvae (*Figure 3A*; *Supplementary file 1*). qPCR results uncovered enriched expression of pro-renin receptor, angiotensinogen, *agtr1a*, *agtr1b*, and Angiotensin Converting Enzyme (*ace*), whereas the expression of *renin* and *ace2* was undetectable in DA neurons compared to the rest of non-DA cells (*Figure 3B*).

To address whether RAAS inhibition is cell autonomously required in DA neurons for neuroprotection, we performed CRISPR-mediated conditional knockout (*Auer et al., 2014*) of *agtr1a* and *agtr1b* in DA neurons (*Figure 3C*). Eight sgRNAs for each gene were designed and screened to identify those with high knockout efficiency, by microinjecting sgRNA and Cas9 protein into one-cell stage embryos followed by sanger sequencing and analysis using the ICE (Inference of CRISPR Edits) software (*Figure 3—figure supplement 1A, B*). The Gal4-UAS system was then used to selectively express the GFP-tagged Cas9 enzyme under the control of *th1* promoter. DNA constructs containing both the UAS-Cas9 and U6 promoter-driven high-efficiency sgRNAs were delivered into one-cell stage *Tg[th1:gal4; uas:NTRmCherry]* embryos, followed by quantification of DA neuronal integrity. The efficacy of *agtr1a* and *agtr1b*-targeting sgRNAs was verified by genotyping DA neurons isolated by manual aspiration (*Figure 3—figure supplement 1C*). We found that DA neurons conditionally expressing Cas9 and effective *agtr1a* and *agtr1b*-targeting sgRNAs (i.e. yellow neurons, expressing Cas9-GFP and NTR-mCherry) were better preserved than those expressing control scrambled sgRNAs following MTZ treatment (*Figure 3D–E*). These results suggest that inhibition of *agtr1* in DA neurons is cell autonomously neuroprotective.

## RAAS inhibitors are neuroprotective for DA neurons in a chemically induced Gaucher disease model and an MPP+ model

Given that NTR-MTZ-induced DA neuron degeneration does not occur in human patient settings, we next tested whether RAAS inhibitors are neuroprotective in other models relevant to human PD. Mutations in the *glucocerebrosidase* (*gba1*) gene cause Gaucher disease (GD), the most common genetic risk factor for PD (*Riboldi and Di Fonzo, 2019*). The zebrafish genetic model for GD has a weak and later-onset DA neurodegeneration phenotype (*Keatinge, 2015*). Chemical inhibition of GBA using conduritol B-epoxide (CBE) has been successfully used to model the disease in both mice (*Vardi, 2016*) and zebrafish (*Artola, 2019*). CBE exhibits some selectivity for GBA1 but can also inhibit lysosomal α-glucosidase (GAA), non-lysosomal glucosylceramidase (GBA2), and lysosomal β-glucuronidase (GUSB).

We first observed that CBE dose-dependently reduced DA neuron integrity and locomotor activity in larval zebrafish, with 500 μM in the medium yielding significant results (*Figure 4—figure supplement 1*). We next tested whether olmesartan exerted a protective effect against CBE-induced DA neuron loss and locomotor deficit. Levodopa was used in comparison. While both levodopa and olmesartan ameliorated locomotor deficits induced by CBE (*Figure 4A–B*), only olmesartan rescued TH immunoreactivity deficits induced by CBE treatment (*Figure 4C–D*). Furthermore, CBE preferentially damaged TH neurons as revealed by the double immunofluorescent staining of TH and 5HT (serotonin) (*Figure 4C and E*).

We also tested whether RAAS inhibitors might protect against the neurotoxin and mitochondrial complex I inhibitor 1-methyl-4-phenyl-1,2,3,6-tetrahydropyridine (MPTP). MPTP is a prodrug to MPP+, and both have been shown to damage DA neurons in larval zebrafish (*Bretaud et al., 2004*; *Lam et al., 2005*). Zebrafish were therefore treated with 1 mM MPP+ (dissolved in 0.02 % Tween-80 to facilitate membrane penetration) from 1 to 3 dpf, and RAAS inhibitors or vehicle (0.02 % DMSO) were administered 4 hrs prior to MPP+ treatment. Imaging of ventral forebrain DA neurons showed

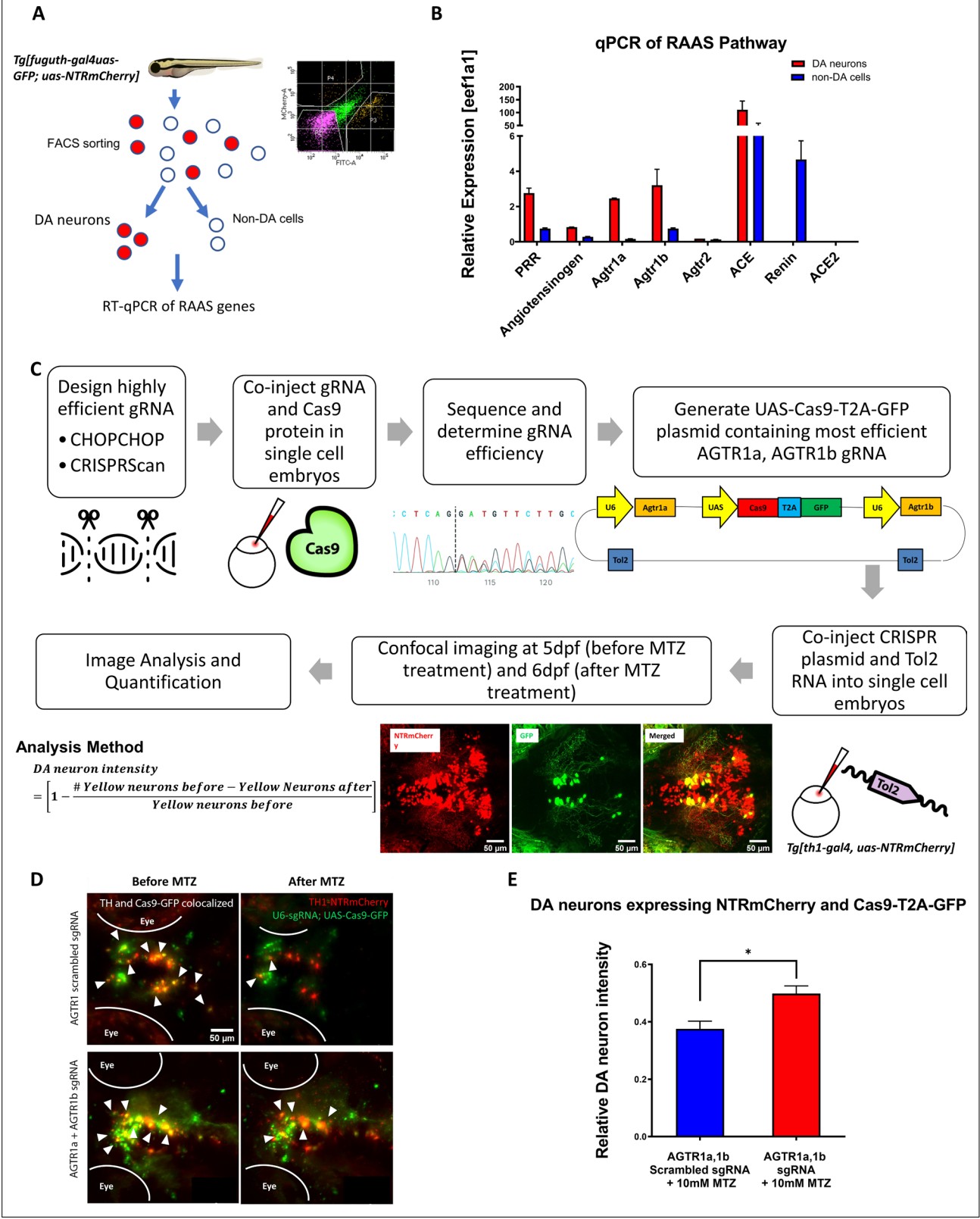

**Figure 3.** Genetic inactivation of *agtr1*a and *agtr1*b in DA neurons is neuroprotective. (**A**) Schematic showing the procedure of FACs to isolate DA neurons for qPCR analysis of RAAS pathway gene expression. (**B**) qPCR data of 5 dpf larval samples show the relative expression of RAAS pathway genes normalized to the house-keeping gene *eef1a1*, in DA neurons (red bars) versus non-DA cells (blue bars). *PRR* (prorenin receptor), *agtr1a* (Angiotensin II receptor, type 1a), *agtr1b* (Angiotensin II receptor, type 1b), *agtr2* (Angiotensin II receptor, type 2), *ace* (Angiotensin I converting enzyme),

*Figure 3 continued on next page*

*Figure 3 continued*

*ace2* (Angiotensin I converting enzyme 2) (n = 2 biological replicates, 6 technical replicates). (**C**) A schematic showing the conditional CRISPR design, imaging, and analysis procedure to inactivate *agtr1a* and *agtr1b* in DA neurons. (**D**) Confocal images of DA neurons in 5 dpf (before MTZ treatment) and 6 dpf (24 hr after 10 mM MTZ treatment) larvae injected with either the scrambled control sgRNA construct (top) or the effective *agtr1a* and *agtr1b* sgRNA construct (bottom). Yellow cells express both NTR-mCherry and Cas9. (**E**) Quantification shows a significant preservation of DA neuron intensity in the *agtr1a* and *agtr1b* sgRNA construct-injected animals compared to the scrambled sgRNA control upon 10 mM MTZ treatment. (n = 15, p < 0.01, unpaired t-test).

The online version of this article includes the following figure supplement(s) for figure 3:

**Source data 1.** DA neuron quantification for conditional CRISPR.

**Source data 2.** Fluorescent activated cell sorting output file.

**Figure supplement 1.** sgRNA design and validation for conditional CRISPR knockout of *agtr1a and 1b* in DA neurons.

that MPP+ treatment significantly reduced DA neuron intensity, and treatment with Olmesartan and captopril significantly protected against DA neuronal loss (*Figure 4—figure supplement 2*).

Together, these results demonstrate that RAAS inhibitors are not only neuroprotective in the synthetic NTR-MTZ model but is also neuroprotective in a Gaucher disease model and a MPP+ model. These findings reinforce the validity of the NTR-MTZ synthetic model for neuroprotective small molecule screening.

## DA neuron-specific RNA-Seq reveals that the AGTR1 inhibitor olmesartan restores the expression of mitochondrial pathway genes disrupted by neurotoxic insults

To further understand the molecular basis underlying the neuroprotective effects of RAAS inhibitors, we carried out DA neuron-specific RNA-seq (*Figure 5A*). *Tg[th1:gal4; uas:NTRmCherry]* larvae were treated with vehicle, CBE, MTZ, olmesartan, CBE+ olmesartan, or MTZ+ Olmesartan for a defined time window, followed by FACs purification of DA neurons from anterior brains and cell type-specific RNA-seq. MTZ and CBE models were theretofore referred to as the neurotoxic models. Upon annotating the sequence reads with the GRCz11 genome assembly, normalizing the read counts, and plotting all the significant gene expression changes ($\alpha$ = 0.05, FDR = 0.1)(*Figure 5—figure supplement 1*), we noted that the two neurotoxic models shared significant overlap and formed distinct clusters compared to the DMSO- or olmesartan alone control groups. Furthermore, treatment of both neurotoxic models with olmesartan restored transcriptomic expression to levels that were similar to controls, especially on the transcriptomes up regulated in the neurotoxic models (*Figure 5B*).

The expression of 1248 genes were commonly altered in the two neurotoxic models compared to vehicle controls (*Figure 5C*), while the expression of 507 genes were commonly altered by olmesartan co-treatment in comparison to each of the neurotoxic insult alone ($\alpha$ = 0.05, FDR = 0.1) (*Figure 5D*). The expression of RAAS pathway genes prorenin receptor (PRR, gene name *atp6ap2*), *agtr1b*, and *ace2* were significantly upregulated in the MTZ treated group compared to the control (padj = 0.001, 0.032, and 0.015 respectively). The *atp6ap2* was also significantly upregulated in the CBE-treated group compared to the control (padj <0.001).

Pathway enrichment analysis with the Reactome and KEGG pathway database showed 28 significantly altered pathways in the neurotoxic models when compared to vehicle controls (p < 0.01). Interestingly, the differentially expressed genes common to both neurotoxic models showed high significance (LogP = –2.55) for the PD KEGG pathway (ID: hsa05012). These results further reinforce the notion that the neurotoxic models used in this study are relevant to PD.

Cluster analysis of the gene ontology and pathways using g:Profiler, DAVID (version 6.8)(*Huang et al., 2009*), and Metascape (*Zhou, 2019*) GO enrichment revealed distinct ontology clusters that were altered in both neurotoxic models compared to controls. Importantly, several pathways related to the mitochondrial function such as ATP synthesis, oxidative stress, and electron transport chain showed the highest significance values (*Figure 5E*). Olmesartan treatment, when compared to the neurotoxic models, significantly affected the clusters related to mitochondrial function, including respiratory electron transport, oxidative phosphorylation, ATP metabolic process, and inorganic cation transport, (*Figure 5F*).

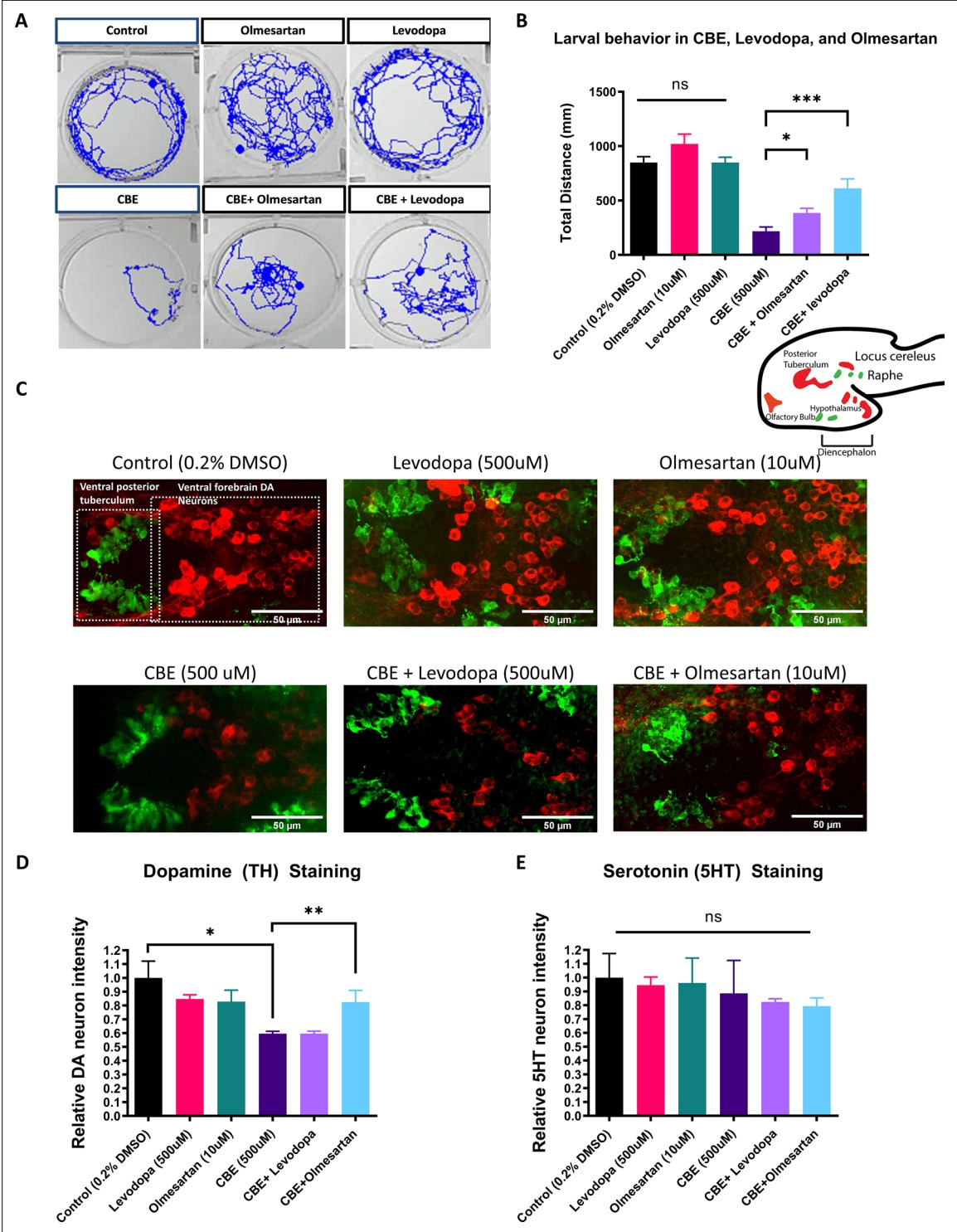

**Figure 4.** The AGTR1 Inhibitor olmesartan is neuroprotective in a chemically induced Gaucher disease model. (**A**) Locomotor tracks of 5 dpf larvae treated 24 hr with 0.2 % DMSO, 500 µM CBE, 10 µM olmesartan, and 500 µM levodopa. The background subtraction method was used to identify and track movement for 5 min duration. (**B**) Quantification of total distance (in millimeters, mm) travelled during 5 min recordings for each sample group. Drugs were added at the indicated concentrations and incubated for 24 hr before behavioral recording (n = 12–13; *p < 0.05, ***p < 0.001, unpaired t test) (**C**) Confocal images of TH-immunoreactive DA neurons (red) and 5HT-immunoreactive serotonin neurons (green) in 6dpf larval zebrafish brains after treatments as indicated in (**B**). (**D–E**) Quantification of neurons in the demarcated regions as shown in (**C**). Fluorescent intensity was quantified using ImageJ and normalized against the control (0.2 % DMSO) (n = 8; *p < 0.05, **p < 0.01, unpaired t test).

The online version of this article includes the following figure supplement(s) for figure 4:

*Figure 4 continued on next page*

*Figure 4 continued*

**Source data 1.** Larval behavior total distance tracking.

**Figure supplement 1.** Dose-dependent effects of CBE on larval zebrafish.

**Figure supplement 2.** Olmesartan and captopril are neuroprotective in a MPP+ larval zebrafish model.

Given the prominence of mitochondrial pathway gene alterations in the neurotoxic models and by olmesartan, we further examined the molecular nature of these genes. As described earlier, preferential mitochondrial DNA damage was observed in DA neurons of the NTR-MTZ model prior to neuronal loss (*Figure 1B*). There is also a strong link to mitochondrial dysfunction in lysosomal storage diseases (*Plotegher and Duchen, 2017*). Thus, disruption of mitochondrial gene expression is possibly causal to DA neuron degeneration in these models.

The differentially expressed genes related to mitochondrial function were further divided into up-regulated and down-regulated categories (*Table 1*). Fourteen genes that were significantly up regulated in the neurotoxic models behaved oppositely upon olmesartan treatment (genes highlighted in blue in *Table 1*). They function in the mitochondrial electron transport chain (e.g. Complex I, III, IV, and V) and TOM-TIM complex critical for protein translocation through the mitochondrial membrane. One gene, *trim3*, which was significantly down-regulated in the neurotoxic models, was up-regulated by olmesartan co-treatment (highlighted in red in *Table 1*). Trim3 (Tripartite motif containing 3), with reported ubiquitin ligase activity, is found to be down-regulated in PD patient plasma (*Dong, 2019*) and can attenuate apoptosis via activating PI3K/AKT signaling pathway in PD models (*Dong, 2020*). Many of these mitochondrial pathways were no longer significantly altered when comparing the olmesartan+ CBE or olmesartan+ MTZ groups to the vehicle control group (*Supplementary file 1*). Taken together, these findings suggest that active AGTR1 receptor is necessary for upregulating the expression of mitochondrial electron transport pathway genes and downregulating *trim3* in both neurotoxic models. Inhibiting its activity can help restore normalcy of these pathways, leading to neuroprotection.

## The AGTR1 inhibitor olmesartan rescues the phenotypes of *Pink1*-deficient *Drosophila*

*Drosophila* offers a plethora of genetic PD models in which DA neuronal loss is evident (*West et al., 2015*; *Lu and Vogel, 2009*). The conserved PINK1-Parkin pathway that directs mitochondrial quality control (MQC) has been originally delineated in flies (*Yang et al., 2006*; *Yang et al., 2003*; *Greene, 2003*; *Park, 2006*; *Clark, 2006*). These models have been used in genetic and pharmacological testing for genes and agents that offer neuroprotection (*Wang, 2006*).

Although the RAAS pathway similar to vertebrates has not been fully described in *Drosophila*, genes encoding the angiotensin converting enzymes are detected in this species (*Coates, 2000*). Recently, it has also been reported that RAAS inhibitors rescue memory defects in a *Drosophila* Alzheimer's disease model (*Lee et al., 2020*). We therefore tested olmesartan in the *Drosophila pink1* model, which recapitulates key features of PD including mitochondrial dysfunction, aberrant mitochondrial morphology, DA neuron and muscular degeneration. The most robust phenotype of the *pink1* mutant flies is the degeneration of their indirect flight muscle caused by the accumulation of dysfunctional and morphologically aberrant mitochondria. This results in flies with collapsed thorax (thoracic indentation) and abnormal wing posture, manifested as droopy or held-up wings as opposed to the straight wings in control animals. Treatment of *pink1* mutant flies by feeding them with food containing 100 μM olmesartan resulted in significant rescue of wing posture (*Figure 6A–C*) and the thoracic indentation (*Figure 6D–F*). The abnormal mitochondrial morphology and neuronal loss phenotypes in DA neurons were also rescued (*Figure 6G–J*). Collectively, these results suggest that olmesartan's protective effect is conserved across species.

## RAAS inhibitors slow down disease progression in human PD patients

Since RAAS inhibitors are commonly used anti-hypertensives, this provided us with an opportunity to ask whether the neuroprotective benefits of RAAS inhibitors shown in zebrafish and *Drosophila* can be observed in human PD patients. We used the Parkinson's Progression Markers Initiative (PPMI) database, which includes a total of 423 de novo PD patients, 308 of which had complete data (accurate

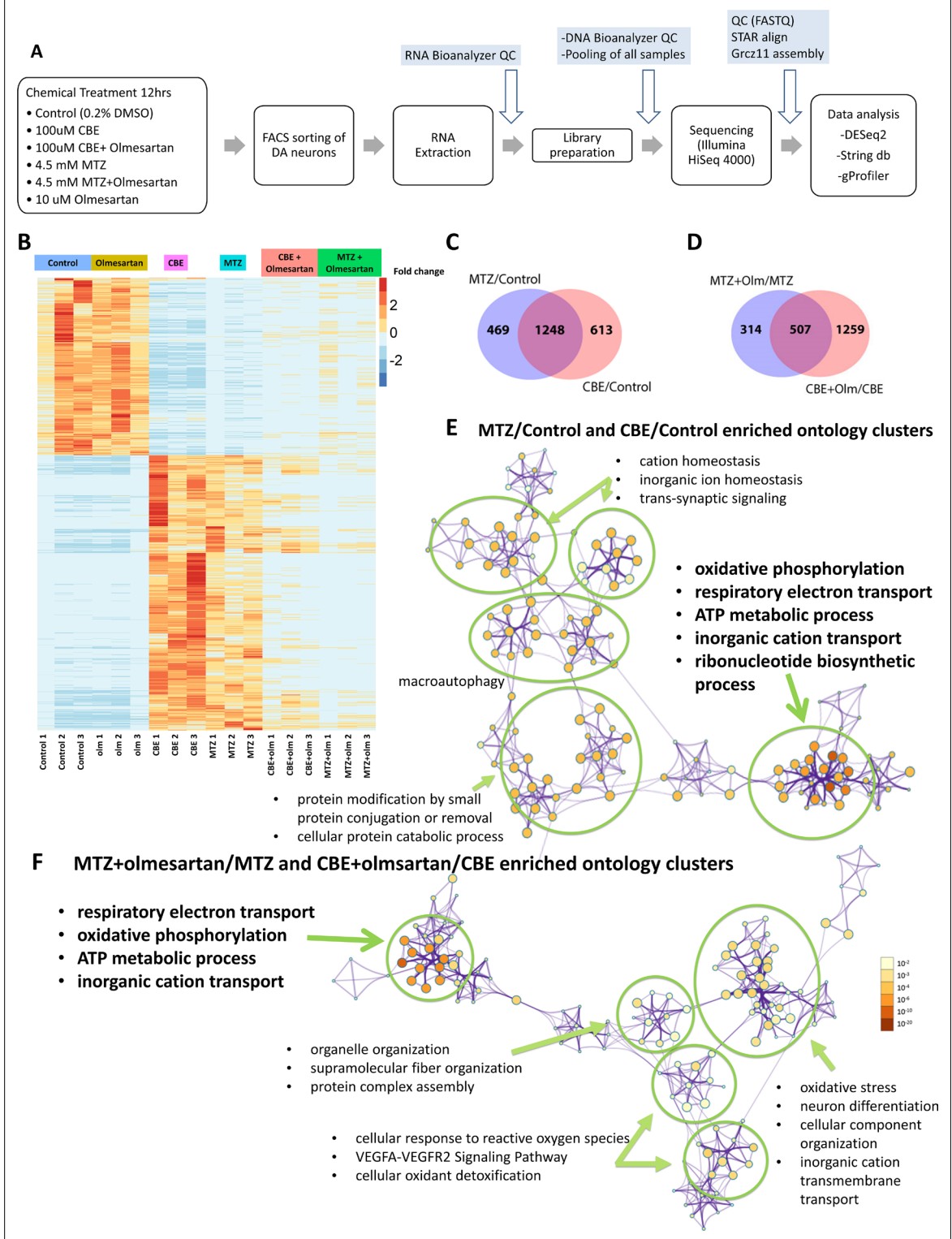

**Figure 5.** DA neuron-specific RNA-seq uncovers neurotoxic insult-induced alterations of mitochondrial pathway gene expression that is in part restored by the AGTR1 inhibitor olmesartan. (**A**) A schematic showing the RNA-seq procedure of larval samples from chemical treatment to FACs, library preparation, and differential gene expression analysis. (**B**) A heatmap of clustering analysis comparing the differential gene expression in DMSO control, olmesartan, CBE, MTZ, MTZ+ olmesartan, and CBE+ olmesartan treatment groups. Gene counts were normalized and analyzed with the R program DESeq2 package. All samples are numbered 1, 2, and 3 to indicate biological replicates. (**C–D**) Venn diagrams showing the overlapping gene expression alterations between different conditions: MTZ/control and CBE/control (**C**) and MTZ+ Olmesartan/MTZ and CBE+ Olmesartan/CBE (**D**) (α = 0.05, FDR

*Figure 5 continued*

= 0.1, Wald test). (**E–F**) Metascape ontology clusters highlighting the top enriched GO terms for differential gene expression common between that of MTZ/Control and CBE/Control (**E**) and that of MTZ+ olmesartan/MTZ and CBE+ olmesartan/CBE (**F**). The colors of the nodes correspond to significant values. The size of the nodes is proportional to the number of input genes in the GO term. The most significant GO terms in both (**E**) and (**F**) include oxidative phosphorylation, respiratory electron transport, ATP metabolic process, and inorganic cation transport.

The online version of this article includes the following figure supplement(s) for figure 5:

**Source data 1.** DESeq2 output with RefSeq (GCF_000002035.6_GRCz11).

**Source data 2.** GO analysis.

**Source data 3.** Cytoscape file for GO analysis CBE/control vs MTZ/control.

**Source data 4.** Cytoscape file for GO analysis CBE+ olm/CBE vs MTZ+ olm/MTZ.

**Figure supplement 1.** Quality control (QC) and pathway analysis of DA neuron-specific RNA-seq data.

**Table 1.** Mitochondrial function-related genes that are commonly up-regulated and down-regulated in the two types of neurotoxic insults (MTZ and CBE), and their significant changes in olmesartan-treated conditions.

Genes highlighted in blue are downregulated in the neurotoxic conditions compared to control but upregulated in olmesartan treatment compared to neurotoxic conditions. Genes highlighted in red are up-regulated in the neurotoxic conditions compared to control, but down-regulated in olmesartan treatment compared to neurotoxic conditions.

| Significantly upregulated mito-expressed genes common in MTZ/C and CBE/C | Significantly downregulated mito-expressed genes common in MTZ/C and CBE/C |
|---|---|
| *Atp5l*: ATP synthase, H + transporting, mitochondrial F0 complex, subunit g (**C-V**) *coa5*: cytochrome c oxidase assembly factor 5 *cox5aa,cox5ab,cox6a1, cox6c, cox7a2a, cox7c, cox8a, mt-co3*: cytochrome c oxidase subunit 5Aa, 5Ab, 6A1, 6 C, 7A2a, 7 C, 8A, III (**C-IV**) *cycsb*: cytochrome c, somatic b (**C-III**) *mdh2*: malate dehydrogenase 2, NAD (mitochondrial) *mrpl13, mrpl32, mrpl36*: mitochondrial ribosomal protein L13, L32, L36 *mrps18c, mrps21*: mitochondrial ribosomal protein s18C, S21 *ndufa1, ndufb2, ndufs4, ndufv2*: NADH:ubiquinone oxidoreductase subunit A1, B2, S4, core subunit V2 (**C-I**) *timm9*: translocase of inner mitochondrial membrane 9 *tomm6, tomm20a, tomm20b*: translocase of outer mitochondrial membrane 6, 20 *uqcc3*: ubiquinol-cytochrome c reductase complex assembly factor 3 *uqcr10*: ubiquinol-cytochrome c reductase, complex III subunit X | *atad3*: ATPase family AAA domain containing 3A *bnip3la*: BCL2 interacting protein three like *cpox:* coproporphyrinogen oxidase *ctnnbip1*: catenin beta interacting protein 1 *kcnh3*: potassium voltage-gated channel subfamily H member 3 *mthfd1l*: methylenetetrahydrofolate dehydrogenase (NADP+ dependent) one like *slc25a37*: solute carrier family 25 member 37 *timm17b*: translocase of inner mitochondrial membrane 17B *top1mt*: DNA topoisomerase I mitochondrial *trim3*: tripartite motif containing 3 |
| **Significantly upregulated mito-expressed genes common in MTZ + Olm/MTZ and CBE + Olm/CBE** | **Significantly downregulated mito-expressed genes common in MTZ + Olm/MTZ and CBE + Olm/CBE** |
| *trim3*: tripartite motif containing 3 *rims3*: regulating synaptic membrane exocytosis 3 *phldb1*: pleckstrin homology like domain family B member 1 | *Atp5l:* ATP synthase, H + transporting, mitochondrial F0 complex, subunit g *cox4i2,cox5aa, cox6a1, cox6c, cox7a2a, cox7c, cox8a:* cytochrome c oxidase subunit 4I2, 5Aa, 6A1, 6 C, 7A2a, 7 C, 8A *cox20*: cytochrome c oxidase assembly factor 20 *cyc1*: cytochrome c-1 *mrpl13,mrpl33, mrpl35, mrpl54:* mitochondrial ribosomal protein L13, L33, L35, L54 *mrps2,mrps18c,mrps36*: mitochondrial ribosomal protein S2, S18C, S36, *ndufa1, ndufb2, ndufs4, ndufv2*: NADH:ubiquinone oxidoreductase subunit A1, B2, S4, core subunit V2 *timm10*: translocase of inner mitochondrial membrane 10 *tomm6*: translocase of outer mitochondrial membrane 6 |

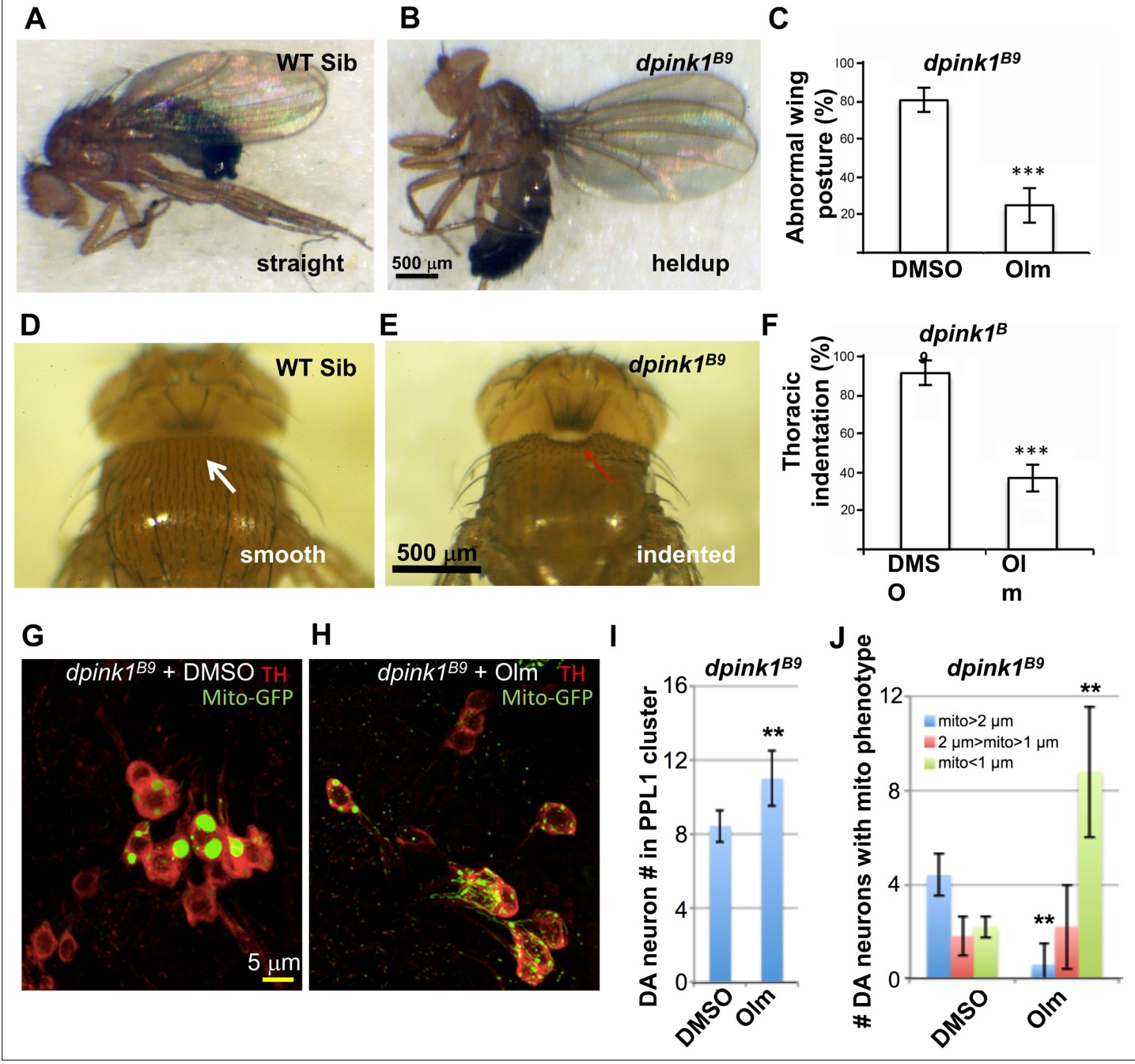

**Figure 6.** The AGTR1 inhibitor olmesartan significantly rescues phenotypes in the *Drosophila pink1* mutant model. (**A–F**) images show the abnormal wing posture (**B**) and thoracic indentation (**E**) in the mutant compared to wild-type siblings (**A,D**). Quantification of %mutant individuals with abnormal wing posture (**C**) and thoracic indentation (**F**) showed a significant difference between vehicle- and drug (olmesartan)-treated samples. (**G–J**) Effect of olmesartan on the mitochondrial aggregation and DA neuron loss phenotypes of *pink1* mutant, in comparison to DMSO control. Mitochondria are labeled with mito-GFP reporter. Data quantification shown in I, J. (n = 12; **, p < 0.01, *** p < 0.001, unpaired t-test).

medication and medical history records for each visit throughout the longitudinal study, no missing records on age, gender, duration of PD, and high visit compliance with no more than three missing records for motor assessment score). The de novo PD patients refer to subjects who have a diagnosis of PD for two years or less and are not taking any PD medications at the time of enrollment. Among them, 96 patients were on RAAS inhibitors (RAAS) while 212 patients were not (non-RAAS). Among the non-RAAS cohort, 42 patients were hypertensive and taking other medications such as calcium channel blockers or diuretics for the management of hypertension (*Figure 7A*).

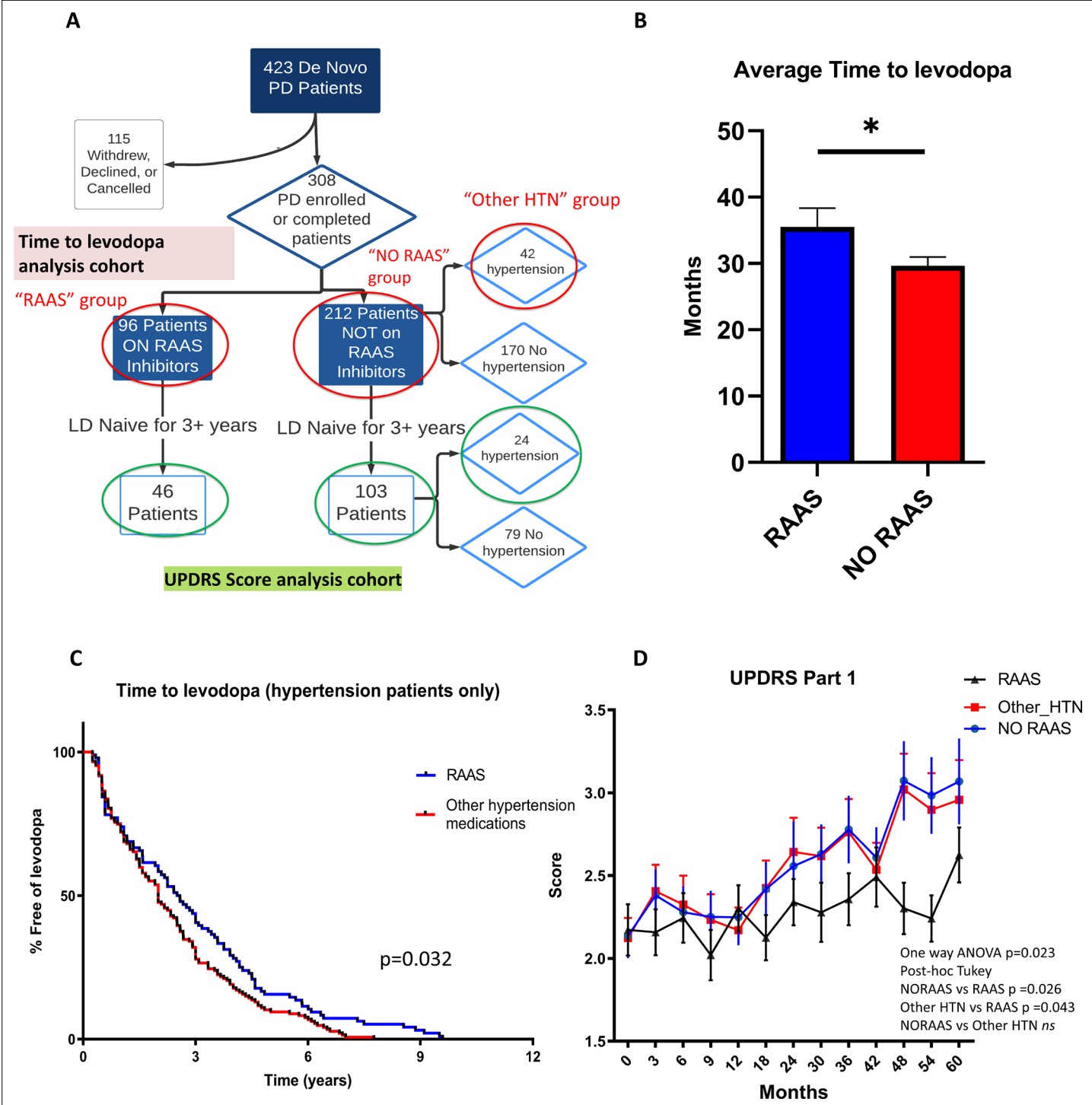

**Figure 7.** Clinical data analysis uncovers delayed disease progression in PD patients on RAAS inhibitors. (**A**) Flow chart showing the patient cohort studied in the PPMI data. Red circles indicate groups of patients on RAAS, not on RAAS, and on other anti-hypertension medications (HTN) used for the time to levodopa analysis. Green circles indicate the patient cohorts not on levodopa for 3+ years that were used for the UPDRS Part 1, 2, and 3 analyses. (**B**) Average time to levodopa therapy for de novo PD patients shows significant difference in patients taking RAAS inhibitors versus patients not on RAAS inhibitors (n = 96 and 212; p < 0.05, unpaired t-test) (**C**) Kaplan Meier survival curve showing the percentage of HTN patients free of levodopa over time for those on RAAS inhibitors versus on other anti-hypertensive medications. HTN patients on RAAS inhibitors showed greater percentage free of levodopa over time compared to patients on other HTN medications (n = 96 and 42; p < 0.05, Log-rank Mantel-cox test). (**D**) UPDRS Score part one shows significantly worsened (higher) scores for subsequent visits in the No RAAS group and the group using other anti-hypertensives compared to the group on RAAS inhibitors (n = 46, 24, and 103; p = 0.023, one-way ANOVA, post-hoc Tukey).

*Figure 7 continued on next page*

*Figure 7 continued*

The online version of this article includes the following figure supplement(s) for figure 7:

**Source data 1.** Parkinson Progression Marker's Initiative deidentified patient data for analysis.

**Source data 2.** UPDRS Score analysis.

**Source data 3.** R script for propensity score matching.

**Source data 4.** PPMI Data and Publications Committee approval letter.

**Figure supplement 1.** Patient cohorts from PPMI data, propensity score matching, and UPDRS Part2, Part3 analysis.

Using this dataset, we sought to compare PD progression in patients on RAAS inhibitors to those who were not. At present, there are no accepted progression biomarkers for PD (*Espay et al., 2017*). The Unified Parkinson's Disease Rating Scale (UPDRS), while widely utilized, suffers from limitations including subjectivity and ambiguities in the written text (*Movement Disorder Society Task Force on Rating Scales for Parkinson's Disease, 2003*). Because of the wear-off and debilitating side effects of levodopa after prolonged use (*Espay, 2018*), clinicians delay the prescription of levodopa to PD patients until absolutely necessary for treating debilitating motor symptoms. Therefore, we used the Time-to-Levodopa as a quantifiable and objective parameter to measure disease progression. After propensity score matching between RAAS and no-RAAS cohorts (Figure S8A-C), our analysis found that the patients on RAAS inhibitors had a significantly delayed onset of levodopa therapy compared to the patients not on RAAS inhibitors (difference, $-5.8$; 95% CI $-11.26$ to $-0.4254$; $p = 0.035$) (*Figure 7B*). To control for hypertension as a variable, we compared patients on RAAS inhibitors to those on other classes of anti-hypertensive medications. This analysis also uncovered a significant effect of RAAS inhibitors in delaying the onset of levodopa therapy as shown in the Kaplan Meier curve ($p = 0.032$) (*Figure 7C*).

The UPDRS scores part I, II, and III were also examined for a subset of patients within each cohort whom were levodopa-naive for at least 3 years; this subset of patients was chosen since levodopa use can significantly influence the UPDRS scores. The UPDRS part I score, which examines the mentation, behavior, and mood, showed a significantly lower score in the cohort on RAAS inhibitors compared to the cohorts not on RAAS inhibitors or those taking other hypertensive medications over the course of 5 years (difference: NO RAAS vs RAAS = 0.289, Other HTN vs RAAS = 0.266, $p = 0.017$) (*Figure 7D*). The UPDRS part 2 and part 3, which examine the activities of daily living and motor function respectively, did not show significance in patients taking RAAS inhibitors when compared to other cohorts ($p = 0.82$) (Figure S8E-F). Taken together, these data suggest that inhibition of RAAS signaling slows down disease progression in human PD patients.

## Discussion

For most neurodegenerative disorders, there exist no disease-modifying therapeutics. Here, we validated the NTR-MTZ-based chemogenetic DA neuron ablation model in zebrafish by showing that DA neuronal loss is preceded by preferential mitochondrial DNA damage and ensuing mitochondrial dysfunction. We then used this system to conduct a high-content DA neuron imaging-based chemical screen and identified the inhibitors of Renin-Angiotensin (RAAS) system to be significantly neuroprotective via cell-autonomous effects of the angiotensin receptor one in DA neurons. Dopamine neuron-specific RNA-seq further revealed the molecular action of RAAS signaling in mitochondrial gene regulation. Finally, inhibition of RAAS signaling was neuroprotective across species. Together, this study identifies RAAS inhibitors as promising therapeutics for slowing down PD progression and highlights a new approach composed of high content screening in zebrafish, cross-species validation, and examination of human clinical data to uncover previously unrecognized neuroprotective agents and underlying mechanisms.

Although MTZ has historically been widely prescribed as an antibiotic, the underlying mechanism of cell death in vertebrates has however remained elusive. By investigating the integrity of mitochondrial and nuclear DNAs, performing live imaging of mitochondria, and DA neuron-specific RNA-seq, we link the mode of NTR-MTZ-mediated cell death to mitochondrial dysfunction for the first time to our knowledge. This property, together with its safe and scalable nature, makes the NTR-MTZ-mediated cell ablation model in larval zebrafish a valuable small molecule discovery tool for disorders where mitochondrial dysfunction is a prevalent underlying pathophysiological mechanism.

Despite the mechanistic relevance of the NTR-MTZ model to PD in the context of mitochondrial dysfunction, this model has several limitations, including not being able to recapitulate the etiology of PD and the time course of neurodegeneration. Given these limitations, it is advisable that additional validation in etiologically relevant models as we have done in this study are to be carried out.

Our unbiased small molecule screen together with cross-species validations has revealed that inhibition of RAAS signaling is significantly neuroprotective for DA neurons in the context of animal PD models and moreover human PD patients. Hits that target different components of the RAAS pathway (e.g. renin, ACE, and *agtr1*) were uncovered from the primary screen and validated in secondary screening. The AGTR1 inhibitor olmesartan was further shown to be neuroprotective in adult zebrafish, in a zebrafish GD model and a MPP+ model, and *Drosophila pink1*-deficient model. The GD model was created by inhibiting the GBA protein with a chemical inhibitor CBE, which preferentially inhibit GBA1 but can also inhibit other related enzymes including GBA2. Despite this limitation, CBE treatment has been previously shown to recapitulate disease phenotypes in both mice and zebrafish (*Vardi, 2016*; *Artola, 2019*). This chemically induced GD model presents several advantages over the genetic model of GD (*Keatinge, 2015*): First, it shows significant DA neuronal loss at larval stages, whereas the genetic GD model only exhibit weak and variable deficit of DA neurons at adult stages. Second, it can be conveniently combined with transgenic lines that label DA neurons. Its conditional nature further allows us to gain temporal control over the access of DA neurons prior to degeneration.

The finding that RAAS inhibitors are also neuroprotective in the *Drosophila pink1*-deficient model suggests that the underlying neuroprotective mechanisms are deeply conserved across evolution. Intriguingly, literature search has uncovered reports of RAAS inhibitors' neuroprotective effects in various animal models of neurodegeneration, but the mechanisms were not well characterized in these studies (*Grammatopoulos, 2007*; *Muñoz, 2006*). Our findings corroborate with these reports and further demystify the actions of RAAS inhibitors, by showing that *agtr1*a and 1b act cell autonomously in DA neurons beyond their conventional actions on the vasculature systems.

The DA neuron-specific RNA-seq has identified mitochondrial pathway genes and *trim3*, the expression of which was perturbed in the neurotoxic models and restored by RAAS inhibition. While the up-regulation of mitochondrial electron transport chain gene expression in the neurotoxic models appears counter-intuitive to the concept of neurodegeneration as a consequence of bioenergetic crisis, it is worth noting that the RNA-seq was carried out prior to overt neurodegeneration. Although upregulation of electron transport chain gene expression could be a compensatory response to mitochondrial damage or dysfunction, it could also lead to dysregulated metabolic pathways resulting in neurodegeneration (*Area-Gomez et al., 2019*). Future experiments to alter the expression of these genes either individually or in combination followed by evaluating DA neuron states will help further verify the cause-effect relationships.

Several lines of evidence suggest that RAAS signaling via active *agtr1* may play a direct role in promoting neurodegeneration via disrupting mitochondrial gene regulation: First, among the pathways that are commonly altered in morphologically intact DA neurons from both neurotoxic models, those that regulate mitochondrial function have the highest significance values. The ~40 mitochondria-related genes were mostly nuclear genes that encode proteins ranging from electron transport chain subunits such as NADH ubiquinone, cytochrome-c oxidase, and ATP synthase to mitochondrial translocation machinery. The notion that olmesartan restored normalcy to a large extent suggests that AGTR1 is the culprit in mediating these gene expression changes. Second, AGTR1 expression is detected in the mitochondria of a variety of cell types including DA neurons (*Valenzuela, 2016*; *Abadir, 2011*), lending support that it may have a direct role in regulating mitochondrial function. Indeed, AGTR1 is reported to promote reactive oxygen species production and activate MAP kinase pathway that leads to the activation of transcription factors including NF-kB and AP-1, and P53 in the context of vascular senescence (*Min et al., 2009*). Intriguingly, significant upregulation of AP-1 was observed in both neurotoxic models, suggesting the following possible model: Neurotoxic insults activate AGTR1, which activates kinase signaling cascades that upregulate the expression of transcription factors such as AP-1, in turn increasing the expression of mitochondrial electron transport chain genes. Such altered mitochondrial gene expression is associated with increased reactive oxygen species production leading to mitochondrial dysfunction and neuronal death. Future follow-up studies on the genes and pathways discovered in this cell type-specific RNA-seq dataset shall provide deeper insights into how active AGTR1 perturbs mitochondrial function and aggravates neurodegeneration.

The evolutionarily conserved actions of RAAS inhibitors together with their prevalent use for anti-hypertension in PD patients prompted us to examine and subsequently discover their significant effect

in slowing down PD progression. Previous studies that interrogate electronic health records (EHR) data reported mixed results regarding the effect of RAAS inhibitors on the incidence of PD (*Lee et al., 2014*; *Warda et al., 2019*). Given the diverse and complex etiology of PD, this is not surprising. In this study, rather than evaluating the incidence of PD, we focused on PD progression using the Time-to-Levodopa therapy as an innovative criterion in addition to the commonly used UPDRS scores. A significant effect of RAAS inhibitors was detected in delaying the time to levodopa therapy. This marker for disease progression can also be applied to other EHR or clinical data where exam metrics are incomplete due to inadequate hospital protocols, or text mining is difficult due to variations in note taking practices by healthcare workers.

Our sample size of 308 PD patients is relatively modest. It would therefore be of interest to further expand this analysis to a larger patient population. It is also worth noting that the blood-brain-barrier (BBB) permeability of currently available RAAS inhibitors in humans vary from compounds to compounds and are generally poor for compounds such as olmesartan and losartan (*Michel et al., 2013*; *Unger, 2003*). This may contribute to the modest neuroprotective effects observed in the clinical data (e.g., UPDRS part 2 and part three scores showing no significant improvements). With expanded patient population, it may be possible to evaluate and compare the BBB profile of RAAS inhibitors and their extent of neuroprotection. Given the cell autonomous mechanisms that we have discovered through animal studies, we postulate that RAAS inhibitors with better BBB penetrating ability will possibly have a higher neuroprotective effect. RAAS pathway components are broadly expressed in the CNS, suggesting that its inhibition could be broadly neuroprotective, not only for PD, but also for other neurodegenerative diseases.

# Materials and methods

## Key resources table

| Reagent type (species) or resource | Designation | Source or reference | Identifiers | Additional information |
|---|---|---|---|---|
| Gene (*Danio rerio*) | *agtr1a* | Ensembl | ENSDART00000021528.7 | |
| Gene (*Danio rerio*) | *agtr1b* | Ensembl | ENSDART00000066834.5 | |
| Gene (*Danio rerio*) | *ace1* | Ensembl | ENSDART00000114637.4 | |
| Gene (*Danio rerio*) | *agt* | Ensembl | ENSDART00000010918.5 | |
| Gene (*Drosophila melanogaster*) | *pink1* | Ensembl | FBtr0100416 | |
| Strain, strain background (*Danio rerio*, AB Wild Type) | AB WT | Zebrafish International Resource Center | ZFIN ID: ZDB-GENO-960809–7 | |
| Strain, strain background (*Danio rerio*, fuguth:gal4-uas:GFP; uas-NTRmCherry) | Tg[fuguth:gal4-uas:GFP; uas-NTRmCherry] | doi:10.1371/journal.pone.0164645 | | |
| Strain, strain background (*Danio rerio*, UAS:mtPAGFP:mtDsRed2) | Tg[UAS:mtPAGFP:mtDsRed2] | doi: 10.1016/jj.nbd.2016.07.020 | ZFIN ID: ZDB-TGCONSTRCT-161116–1 Edward Burton Lab | |
| Strain, strain background (*Danio rerio*, th1:gal4; uas:NTRmCherry) | Tg[th1:gal4; uas:NTRmCherry] | doi: 10.1016/j.nbd.2016.07.020 | Jiulin Du lab | |
| Strain, strain background (*Drosophila melanogaster*) | PINK1[B9] | doi: 10.1038/nature04788 | Gift from the Chung Lab. | |
| Strain, strain background (*Drosophila melanogaster*) | TH-Gal4 | DOI: 10.1002/neu.10185 | Gift from the Birman Lab. | |

*Continued on next page*

*Continued*

| Reagent type (species) or resource | Designation | Source or reference | Identifiers | Additional information |
|---|---|---|---|---|
| Strain, strain background (*Drosophila melanogaster*) | UAS-mito-GFP | doi: 10.1091/mbc.e05-06-0526 | Gift from the Saxton Lab. | |
| Genetic reagent (Morpholino Oligonucleotide) | *agtr1a* | Gene Tools LLC | agtr1a_MO | 0.5 mM (5'-GACGTTGTCCATTTTGGAGA TTTGT-3') |
| Genetic reagent (Morpholino Oligonucleotide) | *agtr1b* | Gene Tools LLC | agtr1b_MO | 0.5 mM (5'-TCATTGCTGATGTTTGGTTCTCCAT-3') |
| Genetic reagent (PCR Master mix) | GoTaq Green Master Mix | Promega | M7122 | |
| Sequence-based reagent | Renin Angiotensin Pathway Primers | This Paper | PCR primers | Refer to **Supplementary file 1** |
| Sequence-based reagent | Nuclear DNA_F | doi: 10.1016/j.ymeth.2010.01.033 | PCR primers | 5'-AGAGCGCGATTGCTGGATTCAC-3' |
| Sequence-based reagent | Nuclear DNA_R | doi: 10.1016/j.ymeth.2010.01.033 | PCR primers | 5'-GTCCTTGCAGGTTGGCAAATGG-3' |
| Sequence-based reagent | Mitochondria DNA_F | doi: 10.1016/j.ymeth.2010.01.033 | PCR primers | 5'-TTAAAGCCCCGAATCCAGGTGAGC-3' |
| Sequence-based reagent | Mitochondria DNA_R | doi: 10.1016/j.ymeth.2010.01.033 | PCR primers | 5'- GAGATGTTCTCGGGTGTGGGATGG –3' |
| Sequence-based reagent | sgRNA primers for agtr1 | This Paper | PCR primers | Refer to **Figure 3—figure supplement 1** |
| Recombinant DNA reagent | agtr1a_1b sgRNA plasmid | This Paper | | Agtr1a and agtr1b sgRNA cloned into addgene: 74,009 |
| Recombinant DNA reagent | Agtr1a_1b sgRNA scrambled | This Paper | | Agtr1a and agtr1b scrambled sgRNA cloned into addgene: 74,009 |
| Peptide, recombinant protein | Cas9-NLS protein | UC Berkeley | https://qb3.berkeley.edu/facility/qb3-macrolab/ | |
| Antibody | anti-AGTR1 (Rabbit polyclonal) | Proteintech | 25343–1-AP | WB(1:500) IF(1:500) |
| Antibody | anti-5HT (Rabbit polyclonal) | Immunostar | cat#20,080 | IF(1:1000) |
| Antibody | anti-TH (Mouse monoclonal) | Immunostar | cat#22,941 | IF(1:500) |
| Antibody | Goat anti-Mouse IgG (H + L) Cross-Adsorbed Secondary Antibody | Thermofisher | Alexa Fluor 488 (cat# A-11001) | IF(1:500) |
| Antibody | Goat anti-Rabbit IgG (H + L) Highly Cross-Adsorbed Secondary Antibody, | Thermofisher | Alexa Fluor 568 (cat# A-11036) | IF(1:1000) |
| Antibody | chicken anti-GFP | Abcam | ab92456 | IF(1:500) |
| Antibody | rabbit anti-TH | Pel-Freez | P41301 | IF(1:1000) |
| Antibody | Goat Anti-Rabbit IgG H&L Horseradish Peroxidase conjugated antibody | Abcam | ab6721 | WB(1:1000) |

*Continued on next page*

*Continued*

| Reagent type (species) or resource | Designation | Source or reference | Identifiers | Additional information |
|---|---|---|---|---|
| Antibody | Rabbit Anti-Mouse IgG H&L Horseradish Peroxidase conjugated antibody | Abcam | ab6728 | WB(1:1000) |
| Commercial assay or kit | Long-Range PCR Kit | QIAGEN | cat# 206,402 | |
| Commercial assay or kit | QuantSeq 3′ mRNA-Seq Library Prep Kit FWD for Illumina | Lexogen | cat# 015 | |
| Commercial assay or kit | QIAprep Gel Extraction Kit | Qiagen | cat# 28,704 | |
| Commercial assay or kit | Mini-PROTEAN TGX Gels | Bio-Rad | cat# 4561083 | |
| Commercial assay or kit | Trans-Blot Turbo Transfer System | Bio-Rad | cat# 1704150 | |
| Chemical compound, drug | Bioactive Compound Library | SelleckChem; curated set from UCSF Small Molecular Discovery Center | cat# L1700 | 10 µM Screening concentration |
| Chemical compound, drug | Olmesartan | Sigma-Aldrich | cat#144689-63-4 | |
| Chemical compound, drug | Aliskiren | Sigma-Aldrich | cat# 62571-86-2 | |
| Chemical compound, drug | Captopril | Sigma-Aldrich | Cat# 173334-58-2 | |
| Chemical compound, drug | Metronidazole | Selleck Chemicals | cat# S1907 | |
| Chemical compound, drug | N-acetylcysteine | Selleck Chemicals | Cat# S1623 | |
| Chemical compound, drug | Conduritol B Epoxide | Sigma Aldrich | Cat#6090-95-5 | |
| Chemical compound, drug | MPP+ | Millipore Sigma | CAS 36913-39-0 | |
| Chemical compound, drug | 1 % low melting point agarose | Sigma Aldrich | Cat#39346-81-1 | |
| Software, algorithm | ImageJ | NIH | RRID:SCR_003070 | 1.5.0 |
| Software, algorithm | Prism 7 | GraphPad | RRID:SCR_002798 | Version 7.03 |
| Software, algorithm | Matlab | Mathworks | RRID:SCR_001622 | Version R2018 |
| Software, algorithm | CellProfiler | The Broad Institute of Harvard and MIT | RRID:SCR_007358 | Version 3.1.8 |
| Software, algorithm | R DESeq2 package | Bioconductor | RRID:SCR_015687 | Version 4.0.1 |
| Software, algorithm | R MatchIt package | | https://github.com/kosukeimai/MatchIt, *Ho et al., 2011* | Version 4.2.0 |
| Software, algorithm | InCell Analyzer | GE Life Sciences | RRID:SCR_015790 | Version 1.9 |
| Software, algorithm | Ethovision XT | Noldus | RRID:SCR_000441 | |
| Software, algorithm | IMARIS | Bitplane | RRID:SCR_007370 | Version 8.4 |
| Software, algorithm | Synthego ICE software | Synthego | https://ice.synthego.com/#/ | version 2.0 |
| Software, algorithm | SnapGene Viewer | SnapGene | RRID:SCR_015053 | |

## Summary

This study was designed to identify neuroprotective small molecules for Parkinson's disease (PD). A chemo-genetic DA neuron degeneration model employing the NTR-MTZ system was first characterized to uncover mitochondrial dysfunction as a plausible cause of cell death. A whole organism DA neuron imaging-based small molecule screen employing such transgenic zebrafish was then carried out. By screening 1,403 bioactive small molecule compounds, the RAAS pathway inhibitors were identified to be significantly neuroprotective. Their neuroprotective actions were further validated in multiple animal models and in human PD patients. Cell type-specific CRISPR and RNA-seq revealed a DA neuron-autonomous regulation of mitochondrial function as a mechanism underlying the neuroprotective effects of RAAS inhibitors. In vivo studies employing zebrafish were approved by the Institutional Animal Care Use Committee at University of California, San Francisco (Approval Number: AN179000). Use of patient data in the PPMI database was approved by the Michael J Fox Foundation PPMI Data and Publications Committee. No statistical methods were used to predetermine sample size. The sample size (*n*) for each experimental group was indicated in the figure legends. The compound treatment, image collection, and data analysis, for the compound screening, manual counting for secondary hit validation of RAAS inhibitors, western blot of morpholino injections, mass spectrometry of adult fish brains, and adult zebrafish behavior studies were performed in a blinded manner. For all other experiments, the investigators were not blinded to allocation during experiments and outcome assessment. All the experiments were replicated at least two independent times.

## Zebrafish husbandry and transgenic lines

Zebrafish were raised on a 14:10 hr light/dark cycle and maintained in the zebrafish facility according to the University of California San Francisco Institutional Animal Care and Use Committee standards. Embryos were raised in Blue Egg Water (2.4 g CaSO4, 4 g IO Salt, 600 μl of 1 % Methylene per 20 L).

The following transgenic lines were used: *Tg[fuguth:gal4-uas:GFP; uas-NTRmCherry]*(for in vivo drug screening, hit validation, MO injection, and behavioral assessment) (*Liu, 2016*); *Tg[UAS:mtPAGFP:mtDsRed2]* for imaging mitochondrial dynamics, kindly provided by Dr. Edward Burton's lab (*Dukes, 2016*); *Tg[th1:gal4; uas:NTRmCherry]*(for CBE double immunofluorescence staining of TH and 5HT, conditional CRISPR knockout of *agtr1a* and *agtr1b*, DA neuron specific RNA-seq). Tg[*th1-gal4*] was kindly provided by Dr. Jiulin Du's lab (*Li, 2015*).

## *Agtr1a* and *agtr1b* morpholino knockdown and western blot validation of knockdown

Morpholino (MO) antisense oligonucleotides that inhibit protein translation were designed for *agtr1a* and *agtr1b* (*Figure 2—figure supplement 3A*) and purchased from Gene Tools, LLC. 0.5 mM *agtr1a* and *agtr1b* MO working solution was mixed with 1 % phenol red and micro-injected into 1 cell stage embryos (estimated 1–4 nls per embryo). At 5 dpf, control and morphants were treated with 9 mM MTZ for 24 hr and confocal imaging was performed with brightfield and DsRed channel at 6dpf. Eight-bit images were cropped to isolate the diencephalic region of the brain and the DA neuron intensity was quantified with ImageJ.

For western blotting, the 6dpf larvae with DMSO or MTZ treatment were collected after performing confocal imaging. Thirty larvae for each group were homogenized in 150 uL of SDS sample buffer and boiled for 10 minutes at 99 °C and transferred to ice. The samples were centrifuged for 1 min at 12,000 rpm and the supernatant was transferred to a new tube with 5 x SDS protein loading buffer. The samples were loaded into Mini-PROTEAN TGX Gels (cat# 4561083) and run at 180 V for 50 min. Transblotting was done using the Trans-Blot Turbo Transfer System (cat# 1704150) and washed with PBS. Primary antibodies were incubated at 4 °C overnight. For the anti-rabbit *agtr1* antibody (Proteintech 25343–1-AP), 1:500 was used; for the anti-mouse beta actin control (Sigma A5441), 1:2000 dilution was used. Horseradish Peroxidase conjugated secondary antibodies were used (Abcam ab6721 and ab6728) with 2 hr incubation. After washing off the secondary antibodies with PBS, the western blot was visualized with the iBright CL750 Imaging System (Invitrogen A44116). The expected bands of 37 kda for anti-beta actin and 50 kda for anti-AGTR1 were identified and analyzed with imageJ using the 'Mean Grey Value' measurement tool.

## In vivo whole organism imaging-based high-content screening assay and secondary validation

The NTR/MTZ model with NTR expressed in DA neurons was used for drug screening, secondary hit validation, and mechanistic studies of DA neuron degeneration. 4.5 mM, 9 mM, or 10 mM MTZ were used in larval zebrafish with varying time courses (ranging from 8 to 48 hr) to achieve different goals (e.g. pre-, mild, moderate, or severe DA neuron loss). Five mM MTZ was used for prolonged treatment in adult zebrafish.

Drug screening was performed in 96-well plates with the bioactive compound library from SelleckChem obtained from the UCSF Small Molecule Discovery Center (SMDC). Ten µM of compounds were dissolved in blue egg water containing 0.2 % DMSO for a total volume of 200 µL. *Tg[fuguth:gal4-uas:GFP; uas:NTRmCherry]* were treated with 200 µM 1-phenyl 2-thiourea (PTU) on 1dpf and at 3dpf, larvae were transferred to 96-well plates containing the screening compounds or 0.2 % DMSO (positive control). Four hr later, 4.5 mM MTZ was added to compound-containing wells as well as wells that serve as negative control. Treatment lasted for 48 hrs. At 5dpf, the larvae were imaged with brightfield and TexRed channels. The multi-pose method (*Liu, 2016*) was used to image DA neurons in vivo using In Cell Analyzer 2000. The images were analyzed using a custom CellProfiler (*Jones, 2008*) pipeline that masks the eyes and auto-detects the DA neurons to calculate the Brain Health Score (BHS) and SSMD score as previously described (*Liu, 2016*). In brief, we compared automated methods for neuronal counting vs. total fluorescence intensity measure, and have found them to be strongly correlated, with the latter revealing the most significant difference between positive control (vehicle-treated) and negative control (MTZ-treated). This is likely because neurodegeneration is often initiated at the level of neuronal processes, followed by the loss of cell bodies. Therefore, we have chosen to quantifying fluorescence intensity in our study.

For secondary hit validations, 40 µL of 1.5 % agarose was added to ensure that the larvae were embedded in a dorsal down position for confocal imaging before and after MTZ treatment. The live confocal imaging was conducted using In Cell Analyzer 6000 with DsRed and FITC channels with 200 ms exposure time. For manual counting of DA neurons, experimenters were blinded to the treatment conditions. Individual larval zebrafish were mounted in 1.5 % agarose. Ventral forebrain DA neurons were observed under a Zeiss epi-fluorescent microscope and counted on one side of the brain in an identical manner across all larval zebrafish. For the dose response studies, concentrations of the RAAS inhibitors were prepared from a series of fold dilutions. The RAAS pathway inhibitors used in the study including olmesartan, aliskiren, captopril, were purchased from Sigma-Aldrich (cat #144689-63-4, 62571-86-2, 173334-58-2). Metronidazole and NAC were purchased from Selleck Chemicals (cat# S1907, S1623). CBE was purchased from Sigma Aldrich (cat# 6090-95-5). MPP+ was purchased from Millipore Sigma (cat # 36913-39-0).

## Adult and larval Zebrafish locomotor behavior assay

For all adult and larval behavior assays, animals were individualized and incubated in their home tanks in a 26 °C behavior room overnight for habituation. Six-well plates were used to house individual larva in each well with 5 mL of total volume per well. The wells were placed on a lightbox and the videos were recorded from a top-down view. For the adult behavior experiments, the fish were individually housed in their home tank with 500 mL of system water. For the 2 -week duration of the adult behavior test, they were fed with flakes in the morning and replaced with fresh water containing the test compounds daily. The recordings were taken from the top view for 5 min. The total distance moved for the 5 -min duration was analyzed through the EthoVision XT software using the dynamic subtraction algorithm with detection limits between 10 and 100 pixels. For larval fish, the static subtraction algorithm was used with detection limits between 10 and 40 pixels.

## Assessment of nuclear and mitochondrial DNA integrity

*Tg[fuguth:gal4-uas:GFP; uas-NTRmCherry]* were treated with PTU (1:100)(200 µM) on 1dpf. At 5dpf, the larvae were treated with 4.5 mM MTZ for 8 hr and immediately transferred to HBSS (Ca/Mg Free) Buffer (Gibco 14170120) and the brains anterior to the mid/hindbrain boundary were acutely dissected and dissociated with TrypLE (Gibco 12604013) for 30 min. DA neurons were collected via mouth pipetting and the genomic DNA was extracted using extraction buffer (10 mM Tris pH 8.2, 10 mM EDTA, 200 mM NaCl, 0.5 % SDS, 200 µg/ml proteinase K). The nuclear DNA was PCR amplified

using the primer sequences: Forward 5' to 3' AGAGCGCGATTGCTGGATTCAC, Reverse 5' to 3' GTCCTTGCAGGTTGGCAAATGG and the mitochondrial DNA was PCR amplified using the primer sequences: Forward 5' to 3' TTAAAGCCCCGAATCCAGGTGAGC, Reverse 5' to 3' GAGATGTTCTCG GGTGTGGGATGG. The target base pair sizes are 10.7 kb and 10.3 kb, respectively. The PCR was performed with the QIAGEN Long-Range PCR Kit (cat# 206402) optimized for long-range amplification of genomic DNA. The PCR was performed with an initial denaturation step at 94 °C for 1 min, 24 cycles (nuclear DNA) or 19 cycles (mitochondrial DNA) of 94 °C for 15 s, 69 °C for 45 s, and 72 °C for 30 s, with final extension at 72 °C for 10 min. The DNA integrity was evaluated by gel electrophoresis (2 % agarose) and the bands were analyzed with ImageJ using the 'Calibrate' function to determine the optical density of the molecular weight standard, the nuclear DNA, and mitochondrial DNA bands.

## In vivo imaging of mitochondrial dynamics

Transgenic zebrafish *Tg[th1:gal4; uas:NTRmCherry]* were crossed with *Tg[UAS:mtPAGFP:mtDsRed2]* and treated with PTU (1:100) (200 µM) at 1dpf. The larvae were screened for *th1-NTRmCherry* on 4dpf and were treated with either 0.2 % DMSO (control) or 4.5 mM MTZ for 16 hr. The larvae were embedded with 1 % low melting point agarose (Sigma 39346-81-1)(1:100 tricaine added, 0.168 µg/mL) in 35 mm glass bottom dishes (Corning). The PA-GFP was activated with the Nikon 40 x WI objective DAPI channel for 1 min. Upon successful activation, the mitochondria were observable under GFP. Live imaging was performed with 10 s intervals for a total of 10 min. The imaging movies were processed with ImageJ and IMARIS software (version 9.7) where the xyz coordinates of the mitochondria movements were obtained. The values were exported to a custom MATLAB script to calculate total displacement, velocity, and direction.

## Mass spectrometry of adult zebrafish for olmesartan detection

Zebrafish were treated with 10 µM of olmesartan medoxomil (Sigma Aldrich cat# 144689-63-4, the pro-drug form of olmesartan) for 14 days. The drug was freshly dissolved in the system water and administered daily. On day 14, adult zebrafish were dissected to collect the body and the brain which were then pooled to obtain approximately 125 mg per sample (n = 10 males, 10 females). Brain samples were prepared by addition of PBS at a 1:5 ratio and homogenizing (Bertin Precellys 24). Homogenates were mixed with acetonitrile and methanol (1:1 v/v) containing 0.05 µg/mL niflumic acid as an internal standard before filtering with Captiva ND plates (0.2 um) into water and analyzed for the active olmesartan (Sigma Aldrich cat# 144689-24-7) with a QTRAP 5500 tandem mass spectrometer (Sciex) coupled with a Nexera X2 series UHPLC (Shimadzu).

## Drug treatment in the chemically induced Gaucher disease model in larval zebrafish

Zebrafish treated with CBE (a chemical inhibitor of GBA) and the RAAS inhibitor olmesartan were tested for both locomotor behavior and confocal imaging of DA neurons. Initially, CBE concentrations ranging from 100 µM, 500 µM, and 1 mM were used to treat embryonic and larval zebrafish from 1dpf to 5dpf with fresh compounds dissolved in Blue Egg Water changed daily to determine that 500 µM is the optimal concentration for the study (*Figure 4—figure supplement 1*). Prior to treatment with CBE, olmesartan, or levodopa, 5dpf larvae were embedded in 96-well glass bottom plates with 1.5 % agarose and blue egg water and imaged using a 20 x objective under the InCell 6000 confocal microscope. CBE (500 µM), olmesartan (10 µM) or levodopa (500 µM) were added to the agarose-embedded larvae. Twenty-four hr later, the 6 dpf larvae were again imaged and the before vs. after TH intensity was quantified using ImageJ. For behavioral recording, 5 dpf larvae were treated with CBE, olmesartan, or levodopa for 24 hr in six-well plates and behavior was analyzed using Ethovision XT using the methodologies described above.

## Drug treatment in the MPP+ model in larval zebrafish

Treatment conditions were used similar to what has been previously described (*Lam et al., 2005*). In brief, 10 µM olmesartan in 0.02 % DMSO was administered to 1 dpf zebrafish. Four hr later, 1 mM MPP+ in 0.02 % Tween-80 was added. The treatment lasted till 3 dpf. Positive control was treated with vehicle only (0.02 % Tween-80, 0.02 % DMSO), and negative control was treated with MPP+ only. DA neurons were imaged and quantified as described above.

## Drug treatment in the *Drosophila Pink1*-deficient model

Newly eclosed *PINK1^{B9}; TH-Gal4> UAS-mito-GFP* male flies were raised on instant fly food (Carolina) or instant fly food containing 100 µM olmesartan. Flies were transferred to fresh vials daily. After two weeks, the flies were scored for wing posture or examined under dissecting microscope for thoracic indentation. Afterwards, flies were dissected for DA neuron staining. At least seven individuals were examined for each condition. Dissected brain tissue samples were briefly washed with 1 x PBS and fixed with 4 % formaldehyde in 1 x PBS containing 0.25 % Triton X-100 for 30 min at room temperature. Fixed samples were subsequently blocked with 1 x PBS containing 5 % normal goat serum and incubated for 1 hr at room temperature followed by incubation with primary antibodies at 4 °C overnight. The primary antibodies used were: chicken anti-GFP (1:5000, Abcam), and rabbit anti-TH (1:1000, Pel-Freez). After three washing steps with 1 x PBS/0.25 % Triton X-100 each for 15 min at room temperature, the samples were incubated with Alexa Fluor 594- and Alexa Fluor 488-conjugated secondary antibodies (1:500, Molecular Probes) for 3 hr at room temperature and subsequently mounted in SlowFade Gold (Invitrogen). Samples were observed under a Leica SP8 confocal microscope and fluorescent confocal images were processed using Photoshop.

## FACs, qPCR, and RNA sequencing

FACs was performed on the BD FACSaria III Cell Sorter with 488 nm, 561 nm and 638 nm channels. To ensure high accuracy of cell sorting, DA neurons were sorted with both 488 and 561 nm channels from *Tg[fuguth:gal4-uas:GFP; uas:NTRmCherry]*. The dead cells stained with DAPI (1 ng/mL) were sorted with the 405 nm channel. The FACs-sorted cells were immediately processed for RNA extraction (Ambion) and converted to cDNA for qPCR. Transcript sequences were obtained from the Ensembl genome browser for zebrafish (GRCz11; https://uswest.ensembl.org/Danio_rerio/Info/Index) as shown in column 6. The primers were designed with NCBI primer blast (https://www.ncbi.nlm.nih.gov/tools/primer-blast/) spanning a product length between 70bp to 200 bp while minimizing self 3' complementary score. All primers were validated with gel electrophoresis prior to qPCR. The Ct values were compared relative to e*ef1a1* as a housekeeping Gene (*Supplementary file 1*).

For RNA-seq, approximately 500 DA neurons were collected per sample with biological triplicates using the FACs procedure described above. RNA was extracted using the Lexogen SPLIT RNA extraction kit (cat 008) and the quality was assessed in the Agilent 2,100 Bioanalyzer (cat# G2939BA). The library was prepared using the Lexogen QuantSeq 3' mRNA-Seq Library Prep Kit FWD for Illumina (cat# 015). The libraries were quality controlled using Agilent 2100 Bioanalyzer and pooled at 20 mL of 3 ng/mL concentration. The RNA-seq was performed on the Illumina HiSeq 4000 (cat SY-401–4001), with single end 50 bp, generating 350 million reads per lane. FastQC was performed for quality check and all sequences showed high per base sequence quality with greater than 75 % uniquely mapped reads aligned against GRCz11 (*Figure 5—figure supplement 1A*). The count normalization was performed using the DESeq2's median of ratios method which accounts for sequencing depth and RNA composition. This normalization method allows for gene count comparisons between samples, which is suitable for comparing differential gene expression across different sample groups with high sensitivity and specificity (*Li et al., 2020*; *Bullard et al., 2010*). To visualize the similarity of our samples, initially a sample-level QC was performed using Principal Component Analysis (PCA) as shown in *Figure 5—figure supplement 1B*. Each dot represents a sample from the respective group. The raw counts for each gene were modeled and the log2 fold changes were shrunken and differential gene expression analysis was performed using DESeq2 in R with an α level of 0.05 and FDR of 0.1. The gene set was then annotated and converted from the zebrafish ensemble Gene (ENSG) to *Homo sapiens* ENTREZ gene ID with gProfiler. The pathway analysis was conducted on DAVID for the KEGG pathway maps and Metascape for enriched ontology clusters.

## Antibody staining of 5HT and DA neurons in larval zebrafish

Antibody staining was performed as previously described (*Yang et al., 2012*). In brief, 6 dpf larvae were fixed in 4 % PFA overnight, washed with PBS, and their brains were dissected. After 24 hr of dehydration in 100 % methanol overnight followed by rehydration, the samples were incubated in primary antibodies for 72 hr at 4 °C with the rabbit anti-5HT (Immunostar cat#20080) and mouse anti-TH (Immunostar cat#22941) primary antibodies. The brains were then subjected to secondary antibody labeling, using Alexa Fluor 488 anti-mouse (cat A-11001) and Alexa Fluor 568 anti-rabbit (cat

A-11036). The brains were stored in 75 % Glycerol and mounted for confocal imaging. The confocal imaging was taken on the Nikon Ti inverted fluorescence microscope with CSU-W1 large field of view using Apo LWD 40 x/1.15 water immersion lens under GFP and RFP channels.

## Conditional CRISPR

The sgRNA sequences were predicted and designed based on the CHOPCHOP and CRISPRscan database (*Figure 3—figure supplement 1A*). The sgRNAs were synthesized and co-injected with Cas9-LS protein (UC Berkeley, https://qb3.berkeley.edu/facility/qb3-macrolab/) into one-cell stage embryos. The genomic DNAs were extracted and sequenced. Upon different designs of sgRNAs (eight for each gene), a maximal knockout efficiency of 58% and 65% were obtained for *agtr1a* and *agtr1b* respectively, based on the analysis with the Synthego ICE software (version 2.0, https://ice.synthego.com/#/). The plasmid backbone used for the conditional knockout construct was the Tol2-pUAS:Cas9T2AGFP ;U6:sgRNA1;U6:sgRNA2, in which UAS elements drive the expression of Cas9 and GFP linked via the T2A peptide and two sgRNA targets can be simultaneously used. The BsaI and BsmBI restriction sites were used for the sgRNA target sequence cloning as previously described (*Auer et al., 2014*). After cloning, the obtained plasmid construct was co-injected with Tol2 transposase mRNA (*Kawakami et al., 2000*) into one-cell stage embryos derived from *Tg[th1:gal4; UAS:NTRmCherry]*. To validate successful knockout of the genes, after live imaging of DA neurons under both GFP and RFP channels, the zebrafish brains were dissociated with TrypLE Express for 30 min and mouth-pipetting was used to collect the GFP$^+$NTRmCherry$^+$ DA neurons for PCR and sequencing (*Figure 3—figure supplement 1B*).

## Parkinson's progression markers initiative data analysis

The PPMI is an observational clinical study providing a comprehensive database for clinical, imaging, and biological data. The PPMI repository contains clinical data with subject demographics, comprehensive medication history, UPDRS motor assessments, and non-motor assessments. The data were downloaded and accessed in May 2019. Initially 423 de novo PD patients, defined as subjects with a diagnosis of PD with two years or less who are not taking any PD medications, were identified, in which 115 patients had missing information or withdrew from the study making the total included subjects to be 308 patients. Among the 308 patients, 96 patients were taking either ACE inhibitors or ARBs (RAAS group) while 212 patients were not taking ACE inhibitors or ARBs (no RAAS group). Based on the medication history, the average Time-to-Levodopa was compared between the two groups. Among the 212 patients, 42 patients had a diagnosis of hypertension (ICD code R03.0) and were taking other medications for the management of their hypertension. A total of 170 patients were neither hypertensive nor taking other blood pressure medications. For the patient cohorts, propensity score matching was used to match the covariates including age, gender, race, smoking, caffeine consumption, alcohol consumption, and history of head injury (*Figure 7—figure supplement 1A-C*) between the cohorts taking RAAS inhibitors (RAAS group) and not taking RAAS inhibitors (no RAAS group). For the UPDRS motor assessment analysis, a subset of patients from each cohort not taking levodopa for at least three years from their initial PPMI enrollment were selected to remove the possible confounding effects of levodopa on motor improvement. Part 1 (non-motor experiences of activities of daily living), part 2 (motor experiences of activities of daily living), and part 3 (motor examination) were assessed (*Figure 7—figure supplement 1E, F*).

## Statistical analysis

The imaging data from screening studies and behavior studies were analyzed by unpaired t-test using R program and Graphpad Prism software and expressed as means ± SEM unless otherwise stated. Wilcoxon rank sum test was used for the analysis of the high throughput screening database. Differential gene expression analysis of the RNA-seq data was done using the DESeq2 package in R and the fold changes of gene expressions were evaluated with Wald test at an α of 0.05 and FDR 0.1. Clinical data analysis of the PPMI database on Time-to-Levodopa was conducted with a Log-rank Mantel-cox test; the UPDRS motor scores were analyzed with nonparametric one-way analysis of variance (ANOVA) and post-hoc Tukey Test.

## Acknowledgements

We thank Michael Munchua and Vivian Yuan for excellent animal care, Drs. Jason Gestwicki, Stanley Prusiner, and Guo lab members for helpful discussions, DeLaine Larsen, Kari Herrington and UCSF Nikon imaging center for assistance with imaging and data analysis, and the UCSF Institute for Neuro-degenerative Diseases (IND) for access of In Cell Analyzer 6,000. Funding: This project was supported by NIH R21 NS082938, R01 NS120219, and DoD CDMRP PD170068 (to SG), NIH R01NS084412 and R01AR074875 (to BL), NIH R01AG058742 (to HL), the UCSF Mary Anne Koda-Kimble Seed Award for Innovation and the Luis Zeh Fellowship (to GK). Data used in the preparation of this article were obtained from the Parkinson's Progression Markers Initiative (PPMI) database (http://adni.loni.usc.edudata/). For up-to-date information on the study, visit wwwppmiinfoorg." "PPMI – a public-private partnership – is funded by the Michael J Fox Foundation for Parkinson's Research and funding partners, including: AbbVie, Allergan, Amathus Therapeutics, Avid Radiopharmaceuticals, Biogen, BioLegend, Bristol Myers Squibb, Celgene, Denali, GE Healthcare, Genentech, GlaxoSmithKline (GSK), Golub Capital, Handl Therapeutics, insitro, Janssen Neuroscience, Lilly, Lundbeck, Merck, Meso Scale Discovery, Neurocrine Biosciences, Pfizer, Piramal, Prevail Therapeutics, Roche, Sanofi Genzyme, Servier, Takeda, Teva, UCB, Verily and Voyager Therapeutics.

## Additional information

### Competing interests
Su Guo: co-founder of Cerepeut. The other authors declare that no competing interests exist.

### Funding

| Funder | Grant reference number | Author |
|---|---|---|
| National Institutes of Health | NS082938 | Su Guo |
| Department of Defense | PD170068 | Su Guo |
| National Institutes of Health | NS084412 | Bingwei Lu |
| National Institutes of Health | AG058742 | Hao Li |
| National Institutes of Health | NS120219 | Su Guo |
| National Institutes of Health | AR074875 | Benjamin Tang |

The funders had no role in study design, data collection and interpretation, or the decision to submit the work for publication.

### Author contributions
Gha-Hyun J Kim, Conceptualization, Data curation, Formal analysis, Methodology, Writing - original draft; Han Mo, Harrison Liu, Zhihao Wu, Data curation, Formal analysis, Methodology; Steven Chen, Jiashun Zheng, Methodology; Xiang Zhao, Daryl Nucum, James Shortland, Longping Peng, Data curation; Mannuel Elepano, Benjamin Tang, Steven Olson, Nick Paras, Data curation, Methodology; Hao Li, Michelle R Arkin, Bo Huang, Methodology, Supervision; Adam R Renslo, Marina Sirota, Methodology, Supervision, Writing – review and editing; Bingwei Lu, Methodology, Resources, Supervision, Writing – review and editing; Su Guo, Conceptualization, Funding acquisition, Project administration, Resources, Supervision, Writing – review and editing

### Author ORCIDs
Gha-Hyun J Kim (iD) http://orcid.org/0000-0001-9073-2034
Bo Huang (iD) http://orcid.org/0000-0003-1704-4141
Su Guo (iD) http://orcid.org/0000-0002-7342-0108

### Ethics

All of the animals were handled according to approved institutional animal care and use committee (IACUC) protocols (#AN179000) of the University of California San Francisco.

### Decision letter and Author response

Decision letter https://doi.org/10.7554/eLife.69795.sa1
Author response https://doi.org/10.7554/eLife.69795.sa2

---

## Additional files

### Supplementary files

- Transparent reporting form
- Supplementary file 1. Table S1 and Table S2.

### Data availability

All data generated or analysed during this study are included in the manuscript and supporting files and source data files have been provided.

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
