## [Decision Letter]

**Acceptance summary:**

The reviewers found your chemo-genetic approach to investigating death of dopaminergic neurons highly interesting. Furthermore, they felt a central finding of your study, namely that inhibition of RAAS signaling can protect dopaminergic neurons from death, provides a compelling hypothesis with potential relevance for the treatment of Parkinson's or other neurodegenerative conditions. Given the apparent conservation of RAAS inhibitor protective effects in flies, zebrafish, and possibly in humans, the findings presented in your paper are likely to be of substantial interest and stimulate additional investigation of the role of RAAS signaling in neuronal survival.

**Decision letter after peer review:**

Thank you for submitting your article "A Zebrafish Screen Reveals Renin-Angiotensin System Inhibitors as Neuroprotective via Mitochondrial Restoration in Dopamine Neurons" for consideration by *eLife*. Your article has been reviewed by 3 peer reviewers, one of whom is a member of our Board of Reviewing Editors, and the evaluation has been overseen by Didier Stainier as the Senior Editor. The following individual involved in review of your submission has agreed to reveal their identity: Hugo J Bellen (Reviewer #3).

Essential revisions:

1) It would be important to clarify how hits were defined, how many hits came from the screen, and how the RAAS hits compared to other hits identified by the screen.

2) Further discussion of the relevance of a NTR-MTZ model for human PD is warranted. How similar is the model to what happens in human PD. To what extent are findings relevant, and what are the limitations of this model.

3) TH neurons can be counted very easily in zebrafish larvae. Quantifying fluorescence can be affected by levels of the transgene and size of individual neurons/axons. The neurons in these experiments would need to be manually counted, which should be possible from the images. Figure 2B, H for example.

4) Appropriate literature needs to be cited and referred to, as the majority of the cited sources are over 5 years old. They therefore may be out of date with the field.

5) Since most of the mechanistic conclusions made in this manuscript are built upon the use of the fughth-gal4 uas-GFP; uas-NTRmCherry expression system, it is really important to determine its specificity in DA neurons. If not, they need to address these issues, and/or state why this is not an issue in their opinion.

6) In Figure 4, the specificity of CBE, the GBA1 inhibitor, is a concern. It can also inhibit GBA2 and 3. Please address this issue.

7) Several issues of quality in the data, organization, figure presentation, and writing are highlighted in the reviews and must be addressed.

*Reviewer #1:*

The authors of this manuscript do several things very effectively. The scope of the studies is impressive, spanning characterization of mitochondrial dysfunction in the zebrafish model, a small molecule screen, mechanistic studies of hit compounds, validation in orthogonal models, and even a retrospective analysis of clinical data from PD patients. Even smaller details, like confirming drugs have crossed the blood brain barrier in zebrafish, have been performed carefully, suggesting a patient and determined commitment to understanding whether the central findings are conserved and relevant for human health. Data quality appears to be good, overall, and the thoroughness of the studies gives confidence that the findings are real and consistent.

The implications of the findings are really quite interesting, namely that RAAS inhibition may delay disease progression in PD and possibly in other related neurodegenerative disorders. This idea is not entirely new-there is some evidence for this idea in the literature, including evidence of RAAS inhibitors ameliorating animal models of PD and some evidence of a benefit to RAAS inhibition in PD patients. But this paper brings new credence and interest to the idea in two ways, first by arriving at the idea through a relatively unbiased phenotypic screen, and second by demonstrating that the effects are cell autonomous effects on the DA neurons themselves, not an indirect effect through modulating hypertension or some other physiological factor. Although the manuscript does not definitively address the mechanism of action, evidence of effects on mitochondrial structure and function are thought provoking and provide opportunity for future exploration. This manuscript is likely to stimulate further exploration of the role of the RAAS in DA neuronal survival and PD, including further mechanistic work and clinical investigation.

This excellent manuscript would be improved by attention to several issues:

1. The process for hit-calling is not described. It would be important to clarify how hits were defined, how many hits came from the screen, and how the RAAS hits compared to other hits identified by the screen.

2. Further discussion of the relevance of a NTR-MTZ model for human PD is warranted. How similar is the model to what happens in human PD. To what extent are findings relevant, and what are the limitations of this model.

3. The fact that RAAS inhibitors are also effective in Gaucher Disease is interesting. Will RAAS inhibition prevent loss of other cell types in other neurodegenerative disorders? Is it anti-apoptotic generally? What are the bounds of its ability to be neuroprotective?

4. Why doesn't RAAS inhibition show any benefit for UPDRS part 2 and 3?

*Reviewer #2:*

The authors using a Transgenic Zebrafish model which destroys the DA neurons after the application of a drug in an attempt to model Parkinson's disease. This was used as an in vivo screening tool to identify neuroprotective compounds.

This highlighted inhibitors of RAAS signalling as lead hits. They subsequently tested their lead compound (olmesartan) in additional fish and fly models.

The authors show that, as these compounds (inhibitors of RAAS signalling) are routinely used to treat other conditions, by analysing PD patients who have taken them, they show that inhibitors of RAAS signalling could be slowing patient symptoms. This implies inhibitors of RAAS signalling maybe also neuroprotective in human patients.

The major strengths of this study are using the zebrafish as an in vivo model to screen potentially neuroprotective drugs against. They also confirm their top hits in other in vivo models related to neuronal health and subsequently include a small retrospective study on PD patients.

However a major weakness, are that the main models used for the screen and the validation are not related to Parkinson's. The NTR system used to destroy DA neurons/ effect their viability, would do so in any cell type it is expressed in. Therefore the screen is investigating compounds that protect against MTZ /NTR toxicity and are not related to PD.

However despite these issues, the authors show data demonstrating that the lead compounds they identify are likely working through a rescue of mitochondrial function, through RNA seq experiments from RNA directly taken form DA neurons. As mitochondrial dysfunction is a key features in PD, coupled with the retrospective PD patient study, suggests that inhibitors of RAAS signalling could have therapeutic benefit.

1. The primary model used for the screen, is not a model of PD but actually a model of MTZ/NTR cell death. The results of the screen and its relevance to PD and neuroprotection are difficult to interpret.

2. The primary assay measures total fluorescence of TH neurons as a rapid way for quantifying neuronal health in the DA system in vivo. This is fine for a large screen, but inappropriate for validation experiments. TH neurons can be counted very easily in zebrafish larvae. Quantifying fluorescence can be affected by levels of the transgene and size of individual neurons/axons. The neurons in these experiments would need to be manually counted, which should be possible from the images. Figure 2B, H for example.

3. Appropriate literature needs to be cited and referred to, as the majority of the cited sources are over 5 years old. They therefore maybe out of date with the field.

For example, the opening line of the Results section states that "no currently available models are suitable for neuroprotective screening." and that "genetic models have a weak and variable late onset degeneration phenotype." The review cited here is from 2010, and these statements are incorrect. Since 2010 (and even before then) many zebrafish genetic models of neurodegeneration (including PD) have been characterised which show neuronal loss and screen able phenotypes – including Lopez 2017, Paquet 2009 (Zf Tau models), Flinn et al. 2013, Zhang et al., 2017 (PINK1 linked PD). Many of these studies include successful Zebrafish neuroprotective screens.

4. The paper is written in a way that is difficult to follow. Techniques and strategies are often referred to vaguely, without proper explanation of their function and how they work. When techniques are explained fully, it often happens mid-way through the paper. For example, in the first results paragraph, it is mentioned that DA neurons are isolated, but without reference to how, or any sort of quality control. FACS is eventually mentioned, but this is several sections later.

5. Adult experiments would need to have their DA neurons quantified to confirm neurodegeneration occurs in Tg line at adult stages. Behavioural defects are not a suitable proxy.

6. It is unclear why a tissue specific Crispr was employed, conventional mutants of atgr1a and 1b could have been utilised.

7. Using additional in vivo models to confirm the efficacy of the compounds is admiral but, the models chosen are not PD models. CBE chemical model would model Gaucher's disease and not PD. Furthermore, complete loss of GBA function does produce phenotypes in larval zebrafish, but without causing neuronal loss. As the data shows that using high concentrations of CBE (Figure 4 supp 1) is lethal, the sub lethal doses used in the study likely cause neuronal loss due to off target effects, not related to GBA1 inhibition.

8. It is interesting that the RAAS inhibitors rescue PINK1 *Drosophila* models of Parkinson's disease. But this is the only example of a legitimate Parkinson's model. It would have been helpful to use more appropriate zebrafish models of PD such as PINK1 KO. Making mutants is time consuming, however MPP+ treatments in fish larval are very robust and rapid at producing DA neuronal loss and is routine within the PD field to test potential neuroprotective compounds.

9. Clearly a lot of data has gone into this manuscript. However, although many data sets are of interest, they seem to go off on tangents and are distracting from the wider narrative (mass spec experiments for example). I would recommend streamlining the results by only including only the most relevant data sets in order to make a more cohesive narrative.

*Reviewer #3:*

In this manuscript Kim et al., used a new technology, chemo-genetic nitroreductase-metronidazole (NTR-MTZ)-based dopamine (DA) neuron ablation model in larval zebrafish. They developed this assay to perform an unbiased drug screen of 1403 bioactive compounds. They discovered that multiple Renin-Angiotensin-Aldosteron-System (RAAS) inhibitors show significant neuroprotection against DA neuronal loss. They further show that these RAAS inhibitors show protective effect sin a chemical-induced zebrafish Gaucher disease model (CBE) and a *Drosophila* pink1-deficiecy model, arguing that the neuroprotective effect of RAAS inhibitors is evolutionary conserved. The authors further confirm their finding in animal models of PD and PD patients. RAAS inhibitors are commonly used anti-hypertension drugs. Hence, they examined the Patient Electronic Health Records to compare PD progression in patients who were on RAAS inhibitors treatment to those who were not. They found that PD patients who were on RAAS inhibitors show a longer levodopa-free period than those who were not, supporting their observations in zebrafish and fly PD models that RAAS inhibitors alleviate degenerative phenotypes.

In summary, the authors provide mechanistic evidence through model organisms (zebrafish and fly) and clinical data from PD patients to support the beneficial effect of RAAS inhibitors in PD patients. This is novel, important and of interest to the research and medical community. A major concern relates to the specificity of their fughth-gal4 usa-GFP; uas-NTRmCherry construct in the DA neurons (Concern 1). Since most of the mechanistic conclusions derived in this manuscript are built upon the use of this expression system, it is important to explore its specificity in DA versus non-DA neurons. In addition, this manuscript is not well written and the figures are poorly organized. They try to present too many data in the main figures. There are also many grammatical errors in this manuscript. The authors need to rewrite parts of the manuscript and redesign figures.

Here is a list of concerns.

1. In Figure 1A and 2C, the control images show strong mcherry signal in eyes. However, the MTZ-treated fish does not show mcherry signal in the eyes. Were these two images taken in the same focal plane? Are there TH^+^ cells in zebrafish eyes? How specific is the expression of fughth-gal4 usa-GFP; uas-NTRmCherry construct in the DA neurons? Since most of the mechanistic conclusions made in this manuscript are built upon the use of this expression system, it is really important to determine its specificity in DA neurons. If not, they need to address these issues, and/or state why this is not an issue in their opinion.

2. Figure 1C-H is one of the poorly organized figures. It is very difficult to see the differences in mitochondrial length, as well as the movement, velocity and direction of mitos in these two images. A proper label is required. Additionally, time-lapse images or movies should be included.

3. The "relative dopamine intensity" of the condition "GFP 9mM MTZ" in Figure 1J is much smaller than the same condition in the other figures (Figure 1I, K, L and M). This level is out of the range of the error bar and hence statistically significant. The authors need to explain this discrepancy.

4. In the manuscript the authors mention that the stability of the parkin mRNA is poor. What is the stability of ectopically expressed parkin mRNA? The authors need to include a control to show the levels of parkin in their experimental time points.

5. In Figure 1L, the authors argue that a-Syn + 9mM MTZ does not affect dopamine levels. How about 4.5mM MTZ + a-Syn? In Figure 1M, 4.5mM MTZ + A53T but not 9mM MTZ + A53T show a difference. Maybe a lower concentration of MTZ (4.5mM) + a-Syn will affect dopamine levels. Moreover, the authors use several concentration of MTZ in this manuscript, 4.5mM, 5mM and 9mM. This lack of consistency questions interpretability.

6. In Figure 3B, the authors argue that the expression of the RAAS pathway genes are up-regulated in DA neurons by comparing them to the non-DA neurons. Does MTZ affect the expression of these genes?

7. In Figure 4, the specificity of CBE, the GBA1 inhibitor, is a concern. It can also inhibit GBA2 and 3. Please address this issue. Moreover, the pattern of the highlighted cells shown in Figure 4C looks different in some images. Were these images taken from the same focal plane? Furthermore, the quantification of the 5HT signals does not match the intensity of the presented images. Again, poor data quality.

8. Figure 5 is a busy and very poorly organized figure. The table should be remove from this figure and presented as an independent table. The ontology clusters (Figure 5C-D), are way too small that the authors had to remove the label of nodes. Without such labels makes these clusters are meaningless to the readers. Moreover, does any of these clusters been experimentally confirmed? Why cram so many data in a figure as there are no restrictions in *eLife*???

9. There are no scale bars in Figure 6. The images in Figure 6D-E are blurry. It is very difficult to see the indented thorax phenotype. The label of the y-axis in Figure 6J is wrong. Again, poor figure quality.

If these issues are addressed I am inclined to accept the manuscript.

---

## [Author Response]

Essential revisions:1) It would be important to clarify how hits were defined, how many hits came from the screen, and how the RAAS hits compared to other hits identified by the screen.

We have described the criteria for defining hits (i.e., based on the SSMD scores as we have previously described, see Liu et al., PLOS ONE, 2016, DOI:10.1371/journal.pone.0164645). We have also stated the reasons why RAAS inhibitors are chosen for further study (i.e., multiple hits targeting the same pathway; literature reporting neuroprotective effects yet without clear mechanisms; retrospective clinical data available for PD patients)(See page 6-7, lines 184-208).

2) Further discussion of the relevance of a NTR-MTZ model for human PD is warranted. How similar is the model to what happens in human PD. To what extent are findings relevant, and what are the limitations of this model.

We have added a paragraph to discuss the relevance and limitations of the NTR-MTZ model. In brief, the mitochondrial link makes the model and the findings relevant. The synthetic nature and the unparalleled time course of phenotypic progression are the limitations of the model (see page 15, line 455-458).

3) TH neurons can be counted very easily in zebrafish larvae. Quantifying fluorescence can be affected by levels of the transgene and size of individual neurons/axons. The neurons in these experiments would need to be manually counted, which should be possible from the images. Figure 2B, H for example.

We agree that TH neurons can be manually counted in zebrafish larvae. We have provided a blinded manual count, which showed that RAAS inhibitors protected DA neurons compared to vehicle-treated animals (See Figure 2-supplement 2A).

As the reviewer acknowledged, such manual counting was impractical for the screening purposes. Additionally, the number of neuronal counts is also sensitive to the levels of the transgene (i.e., higher transgene expression will make more neurons become detectable and countable). In our previous paper that established the high content screening pipeline (Liu et al., PLOS ONE, 2016, DOI:10.1371/journal.pone.0164645), we have compared automated methods for neuronal counting vs total fluorescence intensity measure, and have found them to be strongly correlated, with the latter revealing the most significant difference between positive control (vehicle-treated) and negative control (MTZ-treated). This is likely because neurodegeneration is often initiated at the level of neuronal processes. For these reasons, we have chosen to quantifying fluorescence intensity in our study.

We have clarified these points in the revised manuscript (See material methods section, Page 20, line 614-619).

4) Appropriate literature needs to be cited and referred to, as the majority of the cited sources are over 5 years old. They therefore may be out of date with the field.

We have included more up-to-date references in the revised manuscript according to the reviewer’s suggestions (see page 4, line 111-115, and ref. # 20-23).

5) Since most of the mechanistic conclusions made in this manuscript are built upon the use of the fughth-gal4 uas-GFP; uas-NTRmCherry expression system, it is really important to determine its specificity in DA neurons. If not, they need to address these issues, and/or state why this is not an issue in their opinion.

The specificity of this line for labeling ventral forebrain DA neurons has been previously reported (see Liu et al., PLOS ONE, 2016, DOI:10.1371/journal.pone.0164645). This sub-group of DA neurons are anatomically homologous to the mammalian substantia nigra DA neurons and are the neuronal groups examined in our high content screening. We have clarified this point in the revised manuscript (See page 4, line 122-123).

6) In Figure 4, the specificity of CBE, the GBA1 inhibitor, is a concern. It can also inhibit GBA2 and 3. Please address this issue.

We have addressed the relevance and limitation of the CBE model in the revised manuscript (See Page 10, line 293-297; page 15, line 467-468).

7) Several issues of quality in the data, organization, figure presentation, and writing are highlighted in the reviews and must be addressed.

We appreciate the reviewers’ thorough comments on these points and have amended our manuscript according to reviewers’ suggestions.

Reviewer #1:[...]This excellent manuscript would be improved by attention to several issues:1. The process for hit-calling is not described. It would be important to clarify how hits were defined, how many hits came from the screen, and how the RAAS hits compared to other hits identified by the screen.

We appreciate the reviewer’s comment. See our response to the Essential Revisions point #1.

2. Further discussion of the relevance of a NTR-MTZ model for human PD is warranted. How similar is the model to what happens in human PD. To what extent are findings relevant, and what are the limitations of this model.

We appreciate the reviewer’s comment. See our response to the Essential Revisions point #2.

3. The fact that RAAS inhibitors are also effective in Gaucher Disease is interesting. Will RAAS inhibition prevent loss of other cell types in other neurodegenerative disorders? Is it anti-apoptotic generally? What are the bounds of its ability to be neuroprotective?

We appreciate the reviewer’s comment. It remains to be determined whether RAAS inhibition prevents loss of other cell types in other neurodegenerative disorders. The expression of RAAS pathway components is not limited to DA neurons in the nervous system, suggesting that its inhibition could be broadly neuroprotective.

Based on our data and the literature, angiotensin receptor 1 (AGTR1) activation increases the production of reactive oxygen species resulting in mitochondrial damage/dysfunction and neuronal death (apoptotic and possibly other means of cell death such as necroptosis). We are interested in further understanding the underlying mechanisms using zebrafish as a salient model and hope to determine the bounds of its neuroprotective actions in the future. We have clarified these points in the revised manuscript (page 17, 527-529; page 18, line 533-535)

4. Why doesn't RAAS inhibition show any benefit for UPDRS part 2 and 3?

We appreciate the reviewer’s comment. UPDRS part 1 assesses mentation, behavior, and mood. UPDRS part 2 assesses the activities of daily living, and UPDRS part 3 is a motor examination. It is not entirely clear why RAAS inhibition did not show any benefit for UPDRS part 2 and 3. One possibility is that the currently available RAAS inhibitors are not sufficient to improve all aspects of the disease, due to their inadequate capability to cross the blood-brain barrier in humans. We have discussed this point in the revised manuscript (page 17, line 527-529).

Reviewer #2:[…]Major comments1. The primary model used for the screen, is not a model of PD but actually a model of MTZ/NTR cell death. The results of the screen and its relevance to PD and neuroprotection are difficult to interpret.

We appreciate the reviewer’s comment. See our response to the Essential Revisions point #2.

2. The primary assay measures total fluorescence of TH neurons as a rapid way for quantifying neuronal health in the DA system in vivo. This is fine for a large screen, but inappropriate for validation experiments. TH neurons can be counted very easily in zebrafish larvae. Quantifying fluorescence can be affected by levels of the transgene and size of individual neurons/axons. The neurons in these experiments would need to be manually counted, which should be possible from the images. Figure 2B, H for example.

We appreciate the reviewer’s comment. See our response to the Essential Revisions point #3.

3. Appropriate literature needs to be cited and referred to, as the majority of the cited sources are over 5 years old. They therefore maybe out of date with the field.For example, the opening line of the Results section states that "no currently available models are suitable for neuroprotective screening." and that "genetic models have a weak and variable late onset degeneration phenotype." The review cited here is from 2010, and these statements are incorrect. Since 2010 (and even before then) many zebrafish genetic models of neurodegeneration (including PD) have been characterised which show neuronal loss and screen able phenotypes – including Lopez 2017, Paquet 2009 (Zf Tau models), Flinn et al., 2013, Zhang et al., 2017 (PINK1 linked PD). Many of these studies include successful Zebrafish neuroprotective screens.

We appreciate the reviewer’s comment. We have consulted the literature mentioned by the reviewer, and summarized their key points below:

Lopez 2017 Brain: described an A152T Tau zebrafish model, which caused body curvature and motor neuron loss phenotype; tested 3 autophagy up-regulating chemicals as well as the genetic factors involved in autophagy, and saw effects in ameliorating the phenotype.

Paquet 2009 JCI: generated a gal4/uas Tau mutant zebrafish; saw Tau hyper-phosphorylation, motor neuronal defects, and tested GSK3 inhibitors on the model and showed reduced hyper phosphorylation of Tau in vivo.

Flinn et al., 2013 Annals Neurology: Tilling *pink1* mutant zebrafish displayed DA neuronal defects in both larvae and adults but did not progress worse with age. Up-regulation of TIGAR in the mutant, and the knockdown of which rescued DA deficit in the *pink1* mutant.

Zhang et al., 2017 Cell Chemical Biology: Tilling *pink1* mutant showed rotenone-sensitized disruption of touch-evoked escape response (TEER) assay. Using this assay, the authors performed a chemical screen of 727 compounds and identified three structurally related piperazine phenothiazines (antipsychotics) that stimulate autophagy.

We have cited these papers in the revised manuscript as valuable neurodegeneration models in zebrafish. None of these papers reported a high content screen directly quantifying the extent of neuronal loss. The NTR-MTZ DA neuron degeneration assay offers the robustness that is needed to directly screen for neuro-protection via DA neuron imaging. We have clarified these points in the revised manuscript (page 4, line 111-115).

4. The paper is written in a way that is difficult to follow. Techniques and strategies are often referred to vaguely, without proper explanation of their function and how they work. When techniques are explained fully, it often happens mid-way through the paper. For example, in the first results paragraph, it is mentioned that DA neurons are isolated, but without reference to how, or any sort of quality control. FACS is eventually mentioned, but this is several sections later.

We appreciate the reviewer’s comment. We have clarified these technical points in the revised manuscript (see pages 5, line 131-133).

5. Adult experiments would need to have their DA neurons quantified to confirm neurodegeneration occurs in Tg line at adult stages. Behavioural defects are not a suitable proxy.

We appreciate the reviewer’s comment. We have provided DA neuron quantification data in adult stages (See Figure 2—figure supplement 2E-F).

6. It is unclear why a tissue specific Crispr (see minor comments also) was employed, conventional mutants of atgr1a and 1b could have been utilised.

We appreciate the reviewer’s comment. The reason that a tissue-specific CRISPR was employed is to determine whether the neuroprotective effect of RAAS inhibition is cell (i.e., DA neuron)-autonomous, as opposed to the effect of RAAS inhibition on vasculature. We have clarified this point in the revised manuscript (page 9, line 260-272).

7. Using additional in vivo models to confirm the efficacy of the compounds is admiral but, the models chosen are not PD models. CBE chemical model would model Gaucher's disease and not PD. Furthermore, complete loss of GBA function does produce phenotypes in larval zebrafish, but without causing neuronal loss. As the data shows that using high concentrations of CBE (Figure 4 supp 1) is lethal, the sub lethal doses used in the study likely cause neuronal loss due to off target effects, not related to GBA1 inhibition.

We appreciate the reviewer’s comment. Depending on the nature of GBA mutations, Gaucher disease (GD) shows varying severity from early childhood neurodegeneration to PD.

Hence, GBA mutations are the most common genetic risk factor for PD (Riboldi and Di Fonzo, Cells 8, 364, 2019). Chemical inhibition of GBA has been used to model the disease in both mice (Vardi et al., J. Pathol. 239, 496, 2016) and zebrafish (J. Am. Chem. Soc. 141, 4214, 2019). Moderate DA neuronal loss is also observed in older zebrafish (12 wpf, weeks post fertilization) that are genetically deficient for GBA activity (Keatinge et al., Hum. Mol. Genet. 24, 6640, 2015).

In addition to inhibiting GBA1, CBE can also inhibit other GBAs to a less extent (See our response to the Essential Revisions point #6).

8. It is interesting that the RAAS inhibitors rescue PINK1 *Drosophila* models of Parkinson's disease. But this is the only example of a legitimate Parkinson's model. It would have been helpful to use more appropriate zebrafish models of PD such as PINK1 KO. Making mutants is time consuming, however MPP+ treatments in fish larval are very robust and rapid at producing DA neuronal loss and is routine within the PD field to test potential neuroprotective compounds.

We appreciate the reviewer’s comment. We have provided data showing the neuroprotective effect of olmesartan in the MPP+-treated larval zebrafish (See Figure 4—figure supplement 2, page 10, line 306-313).

9. Clearly a lot of data has gone into this manuscript. However, although many data sets are of interest, they seem to go off on tangents and are distracting from the wider narrative (mass spec experiments for example). I would recommend streamlining the results by only including only the most relevant data sets in order to make a more cohesive narrative.

We appreciate the reviewer’s comment. Olmesartan reportedly has a poor capability to cross the blood brain barrier in humans. Therefore, we found it important to determine whether the drug is indeed present in the zebrafish brain. We saw reduced presence in the brain compared to the body, but the amount detected in the brain is sufficient to engage AGTR1 based on the compound Ki. This increased access to the brain in zebrafish compared to mammals is possibly due to the less tight BBB in zebrafish and/or the continuous bath delivery of the drug. These data are therefore critical to corroborate with the observed cell autonomous effect of AGTR1 inhibition in DA neuron protection. We have clarified these points in the revised manuscript (page 8, line 231-234).

Reviewer #3:[…]Here is a list of concerns.1. In Figure 1A and 2C, the control images show strong mcherry signal in eyes. However, the MTZ-treated fish does not show mcherry signal in the eyes. Were these two images taken in the same focal plane? Are there TH^+^ cells in zebrafish eyes? How specific is the expression of fughth-gal4 usa-GFP; uas-NTRmCherry construct in the DA neurons? Since most of the mechanistic conclusions made in this manuscript are built upon the use of this expression system, it is really important to determine its specificity in DA neurons. If not, they need to address these issues, and/or state why this is not an issue in their opinion.

We appreciate the reviewer’s comment. The red fluorescent signal in the eyes is due to pigment-derived autofluorescence, which, for unknown reasons, display variability across individual larval zebrafish. We have clarified this point in the Figure 1 legend (Page 27, line 599).

Regarding the specificity of this line in labeling the DA neurons of interest, please see our response to the Essential Revisions point #5.

2. Figure 1C-H is one of the poorly organized figures. It is very difficult to see the differences in mitochondrial length, as well as the movement, velocity and direction of mitos in these two images. A proper label is required. Additionally, time-lapse images or movies should be included.

We appreciate the reviewer’s comment. We have added arrowheads to demarcate the elongated mitochondria in MTZ-treated sample (Figure 1C). We have also added movies to show the dynamicity of mitochondria (Figure 1-video 1 and 2).

3. The "relative dopamine intensity" of the condition "GFP 9mM MTZ" in Figure 1J is much smaller than the same condition in the other figures (Figure 1I, K, L and M). This level is out of the range of the error bar and hence statistically significant. The authors need to explain this discrepancy.

We appreciate the reviewer’s comment. The data presented in Figure 1I-M were collected with different batches of animals and in some cases different batches of stock MTZ chemical. We do notice batch-to-batch variability in the effects of MTZ, and therefore always included all necessary controls for each set of data.

4. In the manuscript the authors mention that the stability of the parkin mRNA is poor. What is the stability of ectopically expressed parkin mRNA? The authors need to include a control to show the levels of parkin in their experimental time points.

We appreciate the reviewer’s comment. What we referred to is the general stability of mRNAs after microinjection into one-cell stage zebrafish embryos. This is applicable to all mRNAs (not just parkin mRNA). Due to this consideration, we started MTZ treatment earlier and examined DA neurons at ~ 50 hours post fertilization (hpf) in this set of experiments rather than the 5 days post fertilization (dpf) used in our screen and other experiments described in the manuscript. We have clarified this point in the revised manuscript (page 5-6, line 156-160).

5. In Figure 1L, the authors argue that a-Syn + 9mM MTZ does not affect dopamine levels. How about 4.5mM MTZ + a-Syn? In Figure 1M, 4.5mM MTZ + A53T but not 9mM MTZ + A53T show a difference. Maybe a lower concentration of MTZ (4.5mM) + a-Syn will affect dopamine levels. Moreover, the authors use several concentration of MTZ in this manuscript, 4.5mM, 5mM and 9mM. This lack of consistency questions interpretability.

We appreciate the reviewer’s comment. Figure 1M showed that A53T mRNA injection significantly worsened DA neuron integrity in 4.5 mM MTZ-treated animals (milder degeneration) but not in 9 mM MTZ-treated animals, possibly due to a ceiling effect of degeneration in the latter case. We agree with the reviewer that it is highly possible that WT a-syn can also worsen DA neuron degeneration in 4.5 mM MTZ-treated animals, but we did not have that data. We have clarified these points in the revised manuscript (See page 6, line 165-172).

We used 4.5 mM, 9 or 10 mM MTZ in larval zebrafish to achieve moderate vs. severe DA neuron ablation. We used 5 mM MTZ in the prolonged treatment experiment in adult zebrafish. We have clarified these points in the revised manuscript (see Material Methods section page 20, line 598-602).

6. In Figure 3B, the authors argue that the expression of the RAAS pathway genes are up-regulated in DA neurons by comparing them to the non-DA neurons. Does MTZ affect the expression of these genes?

We appreciate the reviewer’s comment. Figure 3B showed detectable expression of RAAS pathway genes in FACs-purified DA neurons. The expression of certain genes (e.g., PRR-prorenin receptor, AGTR1a, AGTR1b, and ACE) appears to be enriched in DA neurons compared to the rest of the cells.

We have examined our DA neuron-specific RNA-seq data. By comparing control versus MTZ-treated samples, we found that prorenin receptor(*atp6ap2*), *agtr1b*, and *ace2* were significantly upregulated in the MTZ treated group compared to the control (padj=0.001, 0.032, and 0.015 respectively)(See page 11, line 333-339).

7. In Figure 4, the specificity of CBE, the GBA1 inhibitor, is a concern. It can also inhibit GBA2 and 3. Please address this issue. Moreover, the pattern of the highlighted cells shown in Figure 4C looks different in some images. Were these images taken from the same focal plane? Furthermore, the quantification of the 5HT signals does not match the intensity of the presented images. Again, poor data quality.

We appreciate the reviewer’s comment. Regarding the utility of CBE, see our response to the Essential Revisions point #6.

Images in Figure 4C were taken from wholemount larval zebrafish brains immuno-stained with TH (red) and 5HT (green) antibodies. The images shown were taken using the Andor Zyla sCMOS camera with a Z-stack of 10 to 14 planes depending on the sample thickness with a Z-step of 3 μm to cover the rostral region of the ventral posterior tuberculum 5HT and ventral forebrain DA neurons. Due to the thickness of the whole brain and variability in tissue flattening during the slide mounting procedure, images from different brains could appear different. In the revised manuscript, we have included the cropped images that consistently cover the same regions across all samples and highlighted the 5HT and DA neuron regions that were used for quantification (Figure 4C).

8. Figure 5 is a busy and very poorly organized figure. The table should be remove from this figure and presented as an independent table. The ontology clusters (Figure 5C-D), are way too small that the authors had to remove the label of nodes. Without such labels makes these clusters are meaningless to the readers. Moreover, does any of these clusters been experimentally confirmed? Why cram so many data in a figure as there are no restrictions in eLife???

We appreciate the reviewer’s comment. We have rearranged the order of Figure 5 for better flow of the paper and making it less busy. The Cytoscape network clusters figures have been enlarged with better labeling of the nodes. As suggested, we have also removed the table from Figure 5 and made it an independent Table (Table 1).

9. There are no scale bars in Figure 6. The images in Figure 6D-E are blurry. It is very difficult to see the indented thorax phenotype. The label of the y-axis in Figure 6J is wrong. Again, poor figure quality.If these issues are addressed I am inclined to accept the manuscript.

We appreciate the reviewer’s comment. We have added the scale bar and improved the image quality to show the indented thorax phenotype better in revised Figure 6.